# Real-world utilization of SARS-CoV-2 serological testing in RNA positive patients across the United States

Carla V. Rodriguez-Watson[1]*, Natalie E. Sheils[2], Anthony M. Louder[3], Elizabeth H. Eldridge[4], Nancy D. Lin[4], Benjamin D. Pollock[5], Jennifer L. Gatz[6], Shaun J. Grannis[6,7], Rohit Vashisht[8], Kanwal Ghauri[1], Gina Valo[9], Aloka G. Chakravarty[9], Tamar Lasky[9], Mary Jung[10], Stephen L. Lovell[10], Jacqueline M. Major[10], Carly Kabelac[3], Camille Knepper[5], Sandy Leonard[11], Peter J. Embi[6,7], William G. Jenkinson[5], Reyna Klesh[11], Omai B. Garner[12], Ayan Patel[13], Lisa Dahm[13], Aiden Barin[13], Dan M. Cooper[13,14], Tom Andriola[13,15], Carrie L. Byington[13], Bridgit O. Crews[16], Atul J. Butte[8,13], Jeff Allen[17]

1 Reagan-Udall Foundation for the FDA, Washington, District of Columbia, United States of America, 2 OptumLabs, Minnetonka, Minnesota, United States of America, 3 Aetion, New York, New York, United States of America, 4 Health Catalyst, Salt Lake City, Utah, United States of America, 5 Mayo Clinic, Rochester, Minnesota, United States of America, 6 Regenstrief Institute, Indianapolis, Indiana, United States of America, 7 Department of Informatics and Health Services Research, Indiana University School of Medicine, Indianapolis, Indiana, United States of America, 8 Bakar Computational Health Sciences Institute, University of California San Francisco, San Francisco, California, United States of America, 9 Office of the Commissioner, U.S. Food and Drug Administration, Silver Spring, Maryland, United States of America, 10 Center for Devices and Radiological Health, U.S. Food and Drug Administration, Silver Spring, Maryland, United States of America, 11 HealthVerity, Philadelphia, Pennsylvania, United States of America, 12 Department of Pathology and Laboratory Medicine, UCLA Medical Center, Los Angeles, California, United States of America, 13 Center for Data-driven Insights and Innovation, University of California Health, Oakland, California, United States of America, 14 Pediatric Exercise and Genomics Research Center, University of California Irvine School of Medicine, Irvine, California, United States of America, 15 Office of Data and Information Technology, University of California, Irvine, California, United States of America, 16 Department of Pathology and Laboratory Medicine, University of California, Irvine, California, United States of America, 17 Friends of Cancer Research, Washington, District of Columbia, United States of America

☯ These authors contributed equally to this work.
* crodriguezwatson@reaganudall.org

**Data Availability Statement:** All relevant data are contained within the paper and its Supporting information files.

## Abstract

### Background

As diagnostic tests for COVID-19 were broadly deployed under Emergency Use Authorization, there emerged a need to understand the real-world utilization and performance of serological testing across the United States.

### Methods

Six health systems contributed electronic health records and/or claims data, jointly developed a master protocol, and used it to execute the analysis in parallel. We used descriptive statistics to examine demographic, clinical, and geographic characteristics of serology testing among patients with RNA positive for SARS-CoV-2.

**Funding:** Financial support for this work was provided in part by a grant from The Rockefeller Foundation (HTH 030 GA-S). BDP, CK, WGJ used funding provided by Yale University-Mayo Clinic Center of Excellence in Regulatory Science and Innovation (CERSI), a joint effort between Yale University, Mayo Clinic, and the U.S. Food and Drug Administration (FDA) (3U01FD005938) (https://www.fda.gov/). JA supported by award number A128219 and Grant Number U01FD005978 from the FDA, which supports the UCSF-Stanford Center of Excellence in Regulatory Sciences and Innovation. AJB was funded by award number A128219 and Grant Number U01FD005978 from the FDA, which supports the UCSF-Stanford Center of Excellence in Regulatory Sciences and Innovation (CERSI). Its contents are solely the responsibility of the authors and do not necessarily represent the official views of the HHS or FDA. The funders had no role in study design, data collection and analysis, decision to publish, or preparation of the manuscript.

**Competing interests:** I have read the journal's policy and the authors of this manuscript have the following competing interests: Natalie E Sheils is an employee of Optum Labs and owns stock in the parent company UnitedHealth group. Anthony M Louder is a paid employee of Aetion and hold Aetion stock options. Nancy D. Lin was an employee of Health Catalyst at the time the work was performed. Jennifer Gatz is a full-time employee of Regenstrief Institute, which provides independent research services to entities including those within the pharmaceutical and medical device industries. Shaun J. Grannis serves as Chief Medical Information Officer for the Indiana Health Information Exchange, and is a founding partner of Uppstroms, LLC. Carly Kabelac is a paid employee of Aetion and hold Aetion stock options. Carrie L. Byington has intellectual property in and receives royalties from BioFire, Inc. She serves as a scientific advisor to IDbyDNA (San Francisco, CA and Salt Lake City, UT). Dr. Byington is on the Board of the Commonwealth Fund. Atul J. Butte is a co-founder and consultant to Personalis and NuMedii; consultant to Samsung, Mango Tree Corporation, and in the recent past, 10x Genomics, Helix, Pathway Genomics, and Verinata (Illumina); has served on paid advisory panels or boards for Geisinger Health, Regenstrief Institute, Gerson Lehman Group, AlphaSights, Covance, Novartis, Genentech, Merck, and Roche; is a shareholder in Personalis and NuMedii; is a minor shareholder in Apple, Facebook, Alphabet (Google), Microsoft, Amazon, Snap, Snowflake, 10x Genomics, Illumina, Nuna Health, Assay Depot (Scientist. com), Vet24seven, Regeneron, Sanofi, Royalty

## Results

Across datasets, we observed 930,669 individuals with positive RNA for SARS-CoV-2. Of these, 35,806 (4%) were serotested within 90 days; 15% of which occurred <14 days from the RNA positive test. The proportion of people with a history of cardiovascular disease, obesity, chronic lung, or kidney disease; or presenting with shortness of breath or pneumonia appeared higher among those serotested compared to those who were not. Even in a population of people with active infection, race/ethnicity data were largely missing (>30%) in some datasets—limiting our ability to examine differences in serological testing by race. In datasets where race/ethnicity information was available, we observed a greater distribution of White individuals among those serotested; however, the time between RNA and serology tests appeared shorter in Black compared to White individuals. Test manufacturer data was available in half of the datasets contributing to the analysis.

## Conclusion

Our results inform the underlying context of serotesting during the first year of the COVID-19 pandemic and differences observed between claims and EHR data sources–a critical first step to understanding the real-world accuracy of serological tests. Incomplete reporting of race/ethnicity data and a limited ability to link test manufacturer data, lab results, and clinical data challenge the ability to assess the real-world performance of SARS-CoV-2 tests in different contexts and the overall U.S. response to current and future disease pandemics.

## Introduction

Coronavirus disease 2019 (COVID-19) is an infectious disease caused by severe acute respiratory syndrome coronavirus 2 (SARS-CoV-2); originally identified in Wuhan, China in December 2019 [1]. In January 2020, COVID-19 was declared a public health emergency in the United States as the disease continued to spread worldwide. As new variants continue to threaten health and well-being across the globe, valid serology tests are needed to support the characterization of immune response—overall and within different subpopulations—to identify effective treatments, prophylaxis, and mitigation strategies [2, 3]. Given the public health emergency, currently *authorized* serologic assays to test for antibodies against SARS-CoV-2 have not undergone the same evidentiary review standards required for the Food and Drug Administration (FDA) *approval* [4, 5]. A collaboration among the US National Cancer Institute, Centers for Disease Control and Prevention (CDC), Biomedical Advanced Research and Development Authority (BARDA), and the Food and Drug Administration (FDA) led to the development of a dataset to compare the performance characteristics of different serological tests that were independently evaluated using sample panels of patients who were positive and negative for SARS-CoV-2 antibodies [6]. However, as the sample size of the dataset is limited, more robust population-based studies on the accuracy of serology tests are needed to support assay selection and implementation, interpretation of seroepidemiologic studies, and estimates of COVID-19 prevalence and immune response [7]. Additionally, given disproportionate infection rates in communities of color [8] and asymptomatic spread and carriage of COVID-19 [9–12], understanding the best use of serologic tests to estimate the true prevalence of disease and immunity is critical to developing sound public health mitigation strategies that serve all communities.

Pharma, Pfizer, BioNTech, AstraZeneca, Moderna, Biogen, Twist Bioscience, Pacific Biosciences, Editas Medicine, Invitae, Doximity, and Sutro, and several other non-health related companies and mutual funds; and has received honoraria and travel reimbursement for invited talks from Johnson and Johnson, Roche, Genentech, Pfizer, Merck, Lilly, Takeda, Varian, Mars, Siemens, Optum, Abbott, Celgene, AstraZeneca, AbbVie, Westat, several investment and venture capital firms, and many academic institutions, medical or disease specific foundations and associations, and health systems. Atul Butte receives royalty payments through Stanford University, for several patents and other disclosures licensed to NuMedii and Personalis. Carla Rodriguez-Watson receives research support from Merck, Novartis, Pfizer, Lilly, Janssen, and AbbVie. She holds minor stock in Gilead.

A critical piece to enable the assessment of real-world performance is the ability to link manufacturer test information, lab results, and patient healthcare data. Despite several initiatives to improve interoperability of healthcare data, there are few incentives to create digital "bridges" enabling public health and research networks to leverage more complete data sets for rapid analysis and discovery [13]. The absence of unique device identifiers (UDIs) for clear and unambiguous identification of specific diagnostic tests; and the limited integration and flow of manufacturer assay information impedes the interpretation of seroepidemiologic studies and estimates of COVID-19 prevalence.

An initial step to address this challenge is to identify which metadata can be captured and explore approaches to transmitting data between the instrument, laboratory information system (LIS), and electronic health record (EHR). Enabling such interoperability would likewise allow us to assess the real-world performance of serological tests and describe results in the context of clinical symptoms. Additionally, disproportionately high infection rates in underserved communities and asymptomatic carriage and spread of SARS-CoV-2 [9, 11] underscore the need for reliable serologic test reporting to accurately estimate disease prevalence and to develop equitable public health mitigation strategies [14, 15]. Recent studies by the Centers for Disease Control (CDC) describe SARS-CoV-2 seroprevalence across the U.S. from convenience samples retrieved from routine blood chemistry [16], and others describe the duration of antibody response [17–20]. However, to our knowledge, few studies characterize the real-world use of serological testing for COVID-19, particularly in the context of symptoms and race [21].

To address these gaps, the Reagan-Udall Foundation for the FDA, in collaboration with the FDA and Friends of Cancer Research. has convened the COVID-19 Evidence Accelerator (EA). The EA is a consortium of leading experts in health systems research, regulatory science, data science, and epidemiology, specifically assembled to analyze health system data to address key questions related to COVID-19. The EA provides a platform for rapid learning and research using a common analytic plan. In May 2020, the EA launched the Diagnostics EA. As part of the Diagnostics EA, we examined patterns of COVID-19 serological testing using real-world data among the different populations and clinical characteristics. Specifically, our objectives were to 1) understand the current state of data interoperability across instrument, laboratory, and clinical data; 2) describe serological testing by demographic, environmental characteristics (e.g., geographic location), baseline clinical presentation, key comorbidities (e.g., diabetes and cardiovascular disease), and bacterial/viral co-infections (e.g., influenza), and 3) assess the timing of serology testing relative to molecular testing date by the characteristics listed above. Characterizing how serology tests were used (including which tests were used, when, and in whom), as well as potential gaps in data, provide an important context to interpret future results to describe diagnostic accuracy.

## Materials and methods

### Study population and setting

A call to participate in this descriptive analysis was put out to the Evidence Accelerator (EA) community. Six health systems answered the call and collaborated on the Diagnostics EA: Aetion and HealthVerity, Health Catalyst, Mayo Clinic, OptumLabs, Regenstrief Institute, and the University of California Health System. Health Catalyst, Mayo Clinic, and the University of California Health System all utilized EHR data from their respective healthcare delivery systems, Regenstrief Institute accessed EHR clinical data from the Indiana health information exchange [22, 23], while Aetion and OptumLabs utilized medical and pharmacy claims, as well as data directly from laboratories. Furthermore, Aetion drew hospital billing data from the

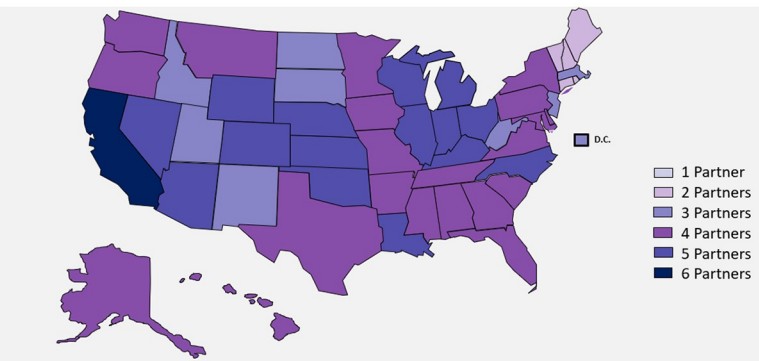

**Fig 1. Geographic coverage of data partners.** Reprinted from brightcarbon.com under a CC BY license, with permission from Bright Carbon, original copyright (2021). Each color represents the number of data partners with a presence in each state but does not necessarily correspond to the number of people. The darkest color represents those where all six partners had a presence.

HealthVerity Marketplace. OptumLabs utilized administrative claims data from a single, large, U.S. insurer. We refer to these health systems as partners A-F for the purposes of anonymity. Data sources included in the analysis are generally categorized as either payer (claims) or healthcare delivery systems. As illustrated in Fig 1, data were drawn from across the U.S. with heavy representation in California, Illinois, Ohio, and Michigan. Characteristics of participating data sources and representative populations are described in S1 Table.

## Study design

Each partner analyzed data collected from their distinct sources according to a master protocol and identified patients across settings (e.g., inpatient, outpatient, or long-term care facility) who tested positive for SARS-CoV-2 ribonucleic acid (RNA) by molecular test between March–September 2020, except one partner who went through April 30, 2021 (Fig 2). "Date of RNA positive" served as the index (cohort entry) date and was defined hierarchically as either the date at 1) sample collection; 2) accession; or 3) result. Among datasets that included primarily claims data, our protocol excluded persons who did not have evidence of enrollment for at least six months in the year before the index to decrease bias in the capture of pre-existing conditions. We did not implement similar data requirements from healthcare delivery systems and health information exchanges (HIEs), given the lack of membership data. We identified comorbidities (pre-existing conditions) 365 days before the index date.

Follow up for serological testing, excluding immunoglobulin M tests, went through 90 days after the index date in all but one partner who identified all RNA positive and serology tests through April 30, 2021 without additional follow-up time for serology. Multiple serological measures were captured. Among those who received a serological test, we described the prevalence of presenting symptoms; concomitant infections with influenza and respiratory syncytial virus; time (in days) to the first serological test; and the number of serological and molecular tests in the 90 days after index.

To minimize the effect of differential missingness between partners, we did the following: 1) included all persons with an office or telephone visit in the +/- 14 days around the index date to enable as complete an assessment of presenting symptoms as possible; 2) in claim systems, included only persons with at least six months of enrollment in the year before index; 3) estimated the proportion of patients at each site who had zero encounters in the prior year to contextualize our capture of pre-existing conditions; and excluded variables from analysis if

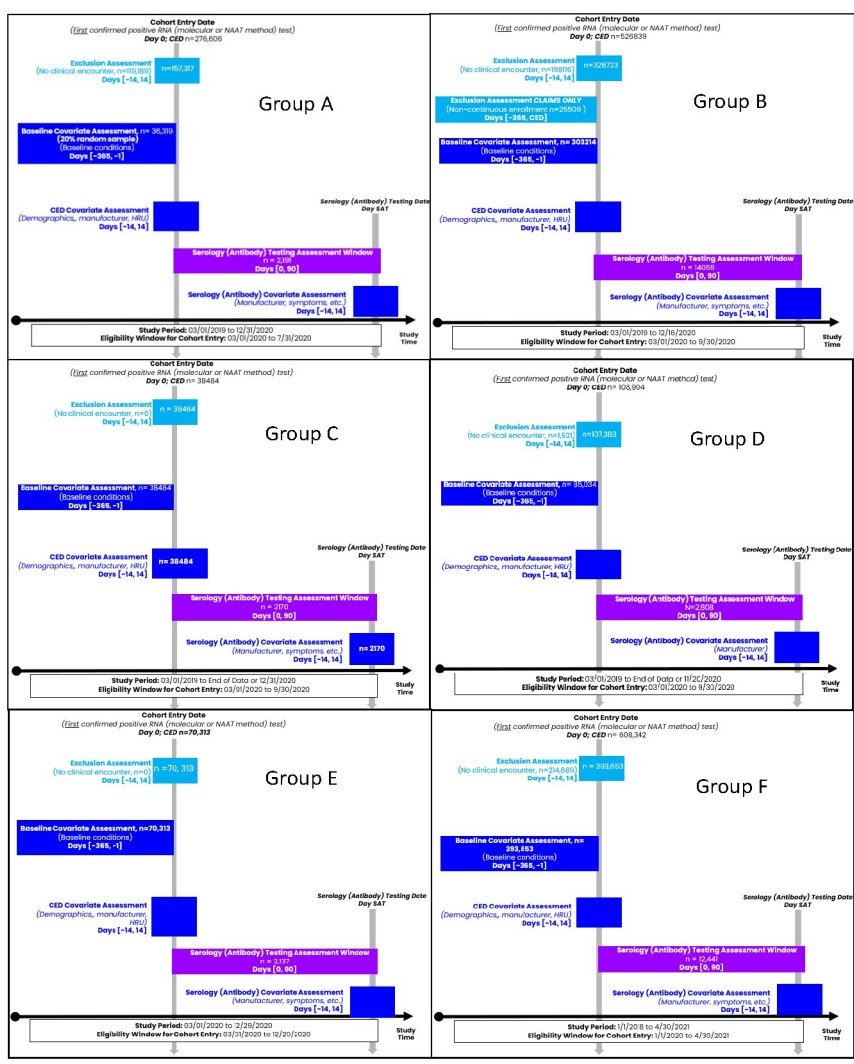

**Fig 2. Study design diagram.**

≥30% of values were missing. Between 35–65% of patients identified from health care delivery systems had no documented encounter in the system in the 365 to 15 days before the index date. In contrast, only 11% of patients from national insurers reported having zero claims in the baseline period. We also assessed the distribution of age, sex, and geography in those with and without data on serology manufacturers. We did not observe any difference by age or sex in those with known versus unknown serology manufacturer information. In a single partner reporting <30% missing race/ethnicity, we observed over-representation of White and Hispanic individuals in those with known serology manufacturer data.

## Measures

Demographic and environmental characteristics, baseline clinical presentation, key comorbidities, bacterial/viral co-infections, and test characteristics potentially related to serological testing were included in the analysis (S1 Fig). We identified comorbidities and clinical presentation using phenotypes defined by ICD-10, and/or National Drug Codes. We provided coding algorithms used for other EA studies and from FDA's Sentinel Initiative for partners to use,

while some partners used existing algorithms generated within their systems. The ICD-10 codes used to identify comorbidities are listed in S2 Table. Given differences in data availability across partners, each partner identified which of the prescribed covariates could be included in their analyses.

## Manufacturer data

We interviewed diagnostic manufacturers, clinical laboratory directors, middleware and information technology vendors, and clients to understand the data generated by the instrument and the data flow from the instrument to information systems for laboratory and clinical data.

## Statistical analysis

Descriptive analyses were performed separately by each contributing data partner in accordance with a common analytic plan. Among persons with and without serology, we calculated the distribution by age, sex, race, ethnicity, U.S. region, pre-existing medical conditions (including but not limited to cardiovascular disease, hypertension, kidney disease, asthma, dementia, chronic liver disease, etc.), smoking status, and obesity. We also analyzed body mass index (BMI), pregnancy status, presenting symptoms, and RNA test manufacturer. Among those with at least one serology test after index, we described the frequency of presenting symptoms and the specific manufacturer/assays at the time of the first serology test, and the time to the first test. We calculated the median and interquartile range (IQR) for the number of days between RNA and the first test. Separately, we included all serology and RNA tests after the index date to describe the median and IQR for the number of molecular and serological tests conducted after the index date.

The WCG Institutional Review Board (IRB), the IRB of record for the Reagan-Udall Foundation for the FDA, reviewed the study and determined it to be non-human subjects research.

## Results

Study samples ranged from 36,319–363,653 individuals per data set—a total of 930,669 people with a confirmed SARS-CoV-2 infection by molecular test across all partners contributing data from March 1- September 30, 2020; and a sixth partner who captured data through April 30, 2020. As described in Table 1, the study population across all datasets was predominantly female, White, and 45–64 years of age. The geographic distribution of patients included in the analyses represented the population in each of the health systems, with two national datasets drawing primarily from the Mid-Atlantic region. Among two datasets, a majority of the sample population had no evidence of pre-existing conditions, whereas in two nationally representative samples, 30–50% had evidence of such. The most prevalent pre-existing conditions across healthcare partners were diabetes, hypertension, cardiovascular disease, obesity, and lung conditions. Across all healthcare partners, 4–11% of the female population were pregnant in the 40 weeks before the index date. The most common presenting symptoms at index were cough, shortness of breath, and pneumonia. The prevalence of lab-confirmed concomitant respiratory syncytial virus or influenza was <1%.

## Serological testing (serotesting)

Generally, 3–6% of those with confirmed infection were serotested–a total of 35,806 people observed.across all datasets. Nearly all follow-up serological tests were immunoglobulin G (IgG) tests (Table 2). Generally, each partner utilized one or two primary serology tests and did not support a large number of tests.

**Table 1. Clinical and demographic characteristics of positive RNA population by serological testing status.**

| Partners | | Total[12] | A | | B | | C | | D | | E | | F | |
|---|---|---|---|---|---|---|---|---|---|---|---|---|---|---|
| | | | N = 36,319 (%) | | N = 303,214 (%) | | N = 38,484 (%) | | N = 85,034 (%) | | N = 70,313 (%) | | N = 393,653 (%) | |
| Serological testing status | | | Yes | No | Yes | No | Yes | No | Yes | No | Yes | No | Yes | No |
| | | | 2,191 (6.0) | 34,128 (94.0) | 14,059 (4.6) | 289,155 (95.4) | 2,170 (5.6) | 36,314 (94.4) | 2,808 (3.3) | 82,226 (96.7) | 2,137 (3.0) | 68,176 (97.0) | 12,441 (3.1) | 381,212 (96.9) |
| Age at time of RNA test[1,2] (Years) | 0–3 | 9,353 (1.0) | 1 (0.0) | 367 (1.1) | 15 (0.1) | 1,919 (0.7) | 5 (0.2) | 750 (2.1) | 0 (0) | 344 (0.4) | 4 (0.2) | 910 (1.3) | 46 (0.4) | 4,992 (1.3) |
| | 4–11 | 28,592 (3.1) | 11 (0.5) | 743 (2.2) | 57 (0.4) | 6,388 (2.2) | 30 (1.4) | 1,266 (3.5) | 13 (0.5) | 2,048 (2.5) | 12 (0.5) | 2,682 (4.0) | 84 (0.7) | 15,258 (4.0) |
| | 12–17 | 46,802 (5.0) | 38 (1.7) | 1,109 (3.0) | 196 (1.4) | 12,049 (4.2) | 40 (1.8) | 1,280 (3.5) | 25 (0.9) | 3,245 (3.9) | 19 (0.9) | 4,042 (5.9) | 213 (1.7) | 24,546 (6.4) |
| | 18–44 | 403,702 (43.5) | 726 (33.2) | 13,012 (38.1) | 5,218 (37.1) | 132,531 (45.8) | 800 (36.9) | 15,222 (41.9) | 834 (29.7) | 39,780 (48.4) | 613 (28.7) | 32,383 (47.5) | 3,341 (26.9) | 159,242 (41.8) |
| | 45–54 | 144,918 (15.6) | 479 (21.9) | 5,811 (17.0) | 2,884 (20.5) | 47,112 (16.3) | 354 (16.3) | 5,530 (15.2) | 484 (17.2) | 12,780 (15.5) | 329 (15.4) | 9,746 (14.3) | 2,184 (17.6) | 57,225 (15.0) |
| | 55–64 | 134,907 (14.6) | 519 (23.7) | 6,047 (17.7) | 2,860 (20.3) | 41,337 (14.3) | 399 (18.4) | 5,422 (14.9) | 579 (20.6) | 10,847 (13.1) | 461 (21.6) | 9,379 (13.8) | 2,564 (20.6) | 54,493 (14.3) |
| | 65–74 | 90,906 (9.8) | 293 (13.4) | 3,702 (10.8) | 1,870 (13.3) | 28,538 (9.9) | 314 (14.5) | 3,816 (10.5) | 479 (17.1) | 6,972 (8.5) | 395 (18.5) | 5,374 (7.9) | 2,264 (18.2) | 36,889 (9.7) |
| | 75–84 | 46,320 (5.0) | 91 (4.2) | 1,929 (5.6) | 761 (5.4) | 13,473 (4.7) | 160 (7.4) | 1,970 (5.4) | 295 (10.5) | 3,997 (4.9) | 212 (9.9) | 2,700 (4.0) | 1,232 (9.9) | 19,500 (5.1) |
| | 85+ | 21,540 (2.3) | 32 (1.5) | 1,432 (4.2) | 198 (1.4) | 5,808 (2.0) | 68 (3.1) | 1,058 (2.9) | 99 (3.5) | 2,213 (2.7) | 92 (4.3) | 960 (1.4) | 513 (4.1) | 9,067 (2.4) |
| Sex[2] | Female | 491,263 (53.0) | 1,274 (58.2) | 19,182 (56.2) | 7,647 (54.4) | 149,668 (51.8) | 1,188 (54.7) | 19,250 (53.4) | 1,531 (54.5) | 43,381 (52.8) | 1,087 (50.9) | 34,546 (50.7) | 7,011 (56.4) | 205,498 (53.9) |
| | Male | 433,733 (46.8) | 891 (40.6) | 13,942 (40.9) | 6,408 (45.6) | 139,419 (48.2) | 982 (45.2) | 17,064 (47.0) | 1,277 (45.5) | 38,810 (47.2) | 1,050 (49.1) | 33,565 (49.2) | 5,429 (43.6) | 174,896 (45.9) |
| | Unknown | 1,790 (0.2) | 18 (0.8) | 782 (2.3) | 4 (0.1) | 68 (0.1) | NA[3] | NA | 0 (0) | 35 (0.1) | 0 (0.0) | 65 (0.1) | <10 (0.0) | 818 (0.2) |
| Race/ Ethnicity[4] | Black | 57,505 (6.2) | NA | NA | 303 (2.2) | 5,993 (2.1) | 83 (3.8) | 2,130 (5.9) | 208 (7.4) | 8,842 (10.7) | 145 (6.8) | 2,718 (4.0) | 1,231 (9.9) | 35,852 (9.4) |
| | White | 470,629 (50.8) | NA | NA | 2,043 (14.5) | 27,879 (9.6) | 1,172 (54) | 16,153 (44.5) | 2,183 (77.7) | 50,960 (62.0) | 1,701 (79.6) | 54,685 (80.2) | 10,173 (81.8) | 303,680 (79.7) |
| | Asian | 13,211 (1.4) | NA | NA | 47 (0.3) | 411 (0.1) | 247 (11.9) | 2,681 (7.4) | 41 (1.5) | 2,003 (2.4) | 88 (4.1) | 1,398 (2.1) | 176 (1.4) | 6,119 (1.6) |
| | Pacific Islander/ Native Hawaiian | 7,585 (0.8) | NA | NA | 2 (0.0) | 39 (0.1) | 6 (0.3) | 254 (0.7) | 13 (0.5) | 909 (1.1) | 3 (0.1) | 66 (0.1) | 110 (1.48) | 6,183 (1.59) |
| | Hispanic or Latino[3] | 103,304 (10.8) | NA | NA | 1,325 (9.4) | 16,134 (5.6) | 685 (31.6) | 12,455 (34.3) | 942 (33.5) | 24,408 (29.7) | 221 (10.3) | 7,197 (10.6) | 1,342 (10.8) | 35,595 (9.3) |
| | American Indian or Alaska Native | 2,916 (0.3) | NA | NA | NA | NA | 7 (0.3) | 151 (0.4) | 94 (3.3) | 1,450 (1.8) | 48 (2.3) | 223 (0.3) | 22 (0.2) | 921 (0.2) |
| | Other | 41,369 (4.5) | NA | NA | NA | NA | 142 (6.5) | 4,518 (12.4) | 65 (2.3) | 2074 (2.5) | NA | NA | 814 (6.5) | 33,756 (8.9) |
| | Missing | 275,156 (29.7) | NA | NA | 10,339 (73.5) | 238,699 (82.5) | 513 (23.6) | 10,427 (28.7) | 40 (1.4) | 5,873 (7.1) | 152 (7.1) | 9,113 (13.4) | NA | NA |

(*Continued*)

**Table 1.** (Continued)

| Partners | | Total[12] | A | | B | | C | | D | | E | | F | |
|---|---|---|---|---|---|---|---|---|---|---|---|---|---|---|
| | | | N = 36,319 (%) | | N = 303,214 (%) | | N = 38,484 (%) | | N = 85,034 (%) | | N = 70,313 (%) | | N = 393,653 (%) | |
| Pre-existing Conditions [5,6] | Cardiovascular disease | 153,329 (16.5) | 924 (43.9) | 13,627 (39.9) | 5,672 (40.3) | 89,722 (31.1) | 1,051 (48.4) | 12,464 (34.3) | 981 (34.9) | 14,175 (17.2) | NA | NA | 891 (6.1) | 13,822 (3.6) |
| | Hypertension | 111,718 (12.1) | 746 (35.4) | 11,710 (36.2) | 4,732 (33.7) | 76,999 (26.6) | 633 (29.2) | 7,675 (21.1) | NA | NA | NA | NA | 412 (3.3) | 8,811 (2.3) |
| | Diabetes | 14,306 (8.9) | 462 (21.9) | 7,023 (19.3) | 2,285 (16.2) | 34,945 (12.1) | 396 (18.2) | 4,835 (13.3) | 484 (17.2) | 7,061 (8.6) | NA | NA | 1,441 (11.6) | 23,159 (6.1) |
| | Cancer | 14,306 (1.5) | 114 (5.4) | 1,342 (4.2) | NA | NA | 268 (12.3) | 2,816 (7.7) | 141 (5.0) | 1,465 (1.8) | NA | NA | 578 (4.6) | 7,582 (2.0) |
| | Asthma | 44,058 (4.8) | 221 (10.5) | 3,085 (9.0) | 1,072 (7.6) | 16,608 (5.7) | 165 (7.6) | 2,636 (7.2) | 191 (6.8) | 3,463 (4.2) | NA | NA | 899 (7.2) | 15,718 (4.1) |
| | Kidney Disease | 35,437 (3.8) | 118 (5.6) | 3,218 (9.4) | 646 (4.6) | 12,589 (4.3) | 437 (20.1) | 4707 (13.0) | 301 (10.7) | 3,586 (4.4) | NA | NA | 682 (5.5) | 9,153 (2.4) |
| | Chronic Lung conditions | 48,880 (5.3) | 297 (14.1) | 5,024 (14.7) | 1631 (11.6) | 27,075 (9.4) | NA | NA | 330 (11.7) | 5,727 (7.0) | NA | NA | 556 (4.5) | 8,240 (2.2) |
| | Auto-Immune conditions | 22,497 (2.4) | 90 (4.1) | 1,056 (3.1) | 1,324 (9.4) | 16,790 (5.8) | 111 (5.11) | 1,084 (3.0) | NA | NA | NA | NA | 122 (1.0) | 1,920 (0.5) |
| | HIV | 1,217 (0.1) | 18 (0.9) | 223 (0.7) | NA | NA | 23 (1.1) | 496 (1.4) | 7 (0.2) | 103 (0.1) | NA | NA | 27 (0.2) | 320 (0.1) |
| | Any liver disease | 16,342 (1.8) | 148 (7.0) | 1,511 (4.7) | 693 (4.9) | 8,246 (2.8) | 223 (10.3) | 2,646 (7.2) | 127 (4.5) | 1412 (1.7) | NA | NA | 87 (0.7) | 1,249 (0.3) |
| | Obesity | 37,388 (4.0) | 510 (23.3) | 7,567 (22.2) | NA | NA | 272 (12.5) | 3,808 (10.5) | 346 (12.3) | 5,765 (7.0) | NA | NA | 881 (7.1) | 18,239 (4.8) |
| | Dementia | 7,240 (0.8) | 25 (1.2) | 1,628 (4.8) | NA | NA | 57 (2.6) | 814 (2.2) | 45 (1.6) | 1294 (1.6) | NA | NA | 118 (0.9) | 3,259 (0.9) |
| | No pre-existing conditions[11] | 547,683 (59.1) | 429 (19.6) | 9,929 (29.1) | 6,666 (47.4) | 170,044 (58.8) | NA | NA | NA | NA | NA | NA | 10,565 (84.9) | 350,050 (91.8) |
| Pregnancy Status Among Females [5,7] | Yes | 5,198 () | 64 (5.0) | 1,199 (6.25) | NA | NA | 148 (12.5) | 2,022 (10.5) | 110 (3.9) | 1,655 (2.0) | NA | NA | NA | NA NA |
| | No | 101,537 () | NA | NA | NA | NA | 1,040 (87.5) | 17,228 (89.5) | 2,698 (96.1) | 80,571 (98.0) | NA | NA | NA | NA |
| Geography [2,8] (patient residence) | New England | 13,057 (1.4) | 80 (3.7) | 2,777 (8.1) | 289 (2.1) | 9,899 (3.4) | NA | NA | NA | NA | 1 (0.0) | 11 (0.0) | 0 | 0 |
| | Mid-Atlantic | 51,390 (5.5) | 1,478 (67.5) | 11,700 (34.3) | 4,757 (33.8) | 33,402 (11.5) | NA | NA | NA | NA | 6 (0.3) | 32 (0.1) | 0 (0) | 15 (0) |
| | South-Atlantic | 84,156 (9.1) | 164 (7.5) | 6,078 (17.8) | 3,853 (27.4) | 68,344 (23.6) | NA | NA | NA | NA | 378 (17.7) | 5,287 (7.8) | 0 (0) | 52 (0) |
| | East North Central | 440,206 (47.5) | 110 (5.0) | 2,845 (8.3) | 524 (3.7) | 34,833 (12.1) | NA | NA | NA | NA | 198 (9.3) | 21,592 (31.7) | 12,202 (98.1) | 367,902 (96.5) |
| | East South Central | 12,614 (1.4) | 12 (0.5) | 488 (1.3) | 257 (1.8) | 10,680 (3.7) | NA | NA | NA | NA | 6 (0.3) | 39 (0.1) | 10 (0.1) | 1,122 (0.3) |
| | West North Central | 62,856 (6.8) | 13 (0.6) | 776 (2.3) | 277 (2.0) | 25,379 (8.8) | NA | NA | NA | NA | 927 (43.4) | 35,472 (52.0) | 0 (0) | 12 (0) |
| | West South Central | 51,389 (5.5) | 92 (4.2) | 4,248 (12.4) | 1,609 (11.4) | 45,249 (15.6) | NA | NA | NA | NA | 12 (0.6) | 162 (0.2) | <10 (0) | 17 (0) |
| | Mountain | 55,782 (6.0) | 26 (1.2) | 544 (1.6) | 1,952 (13.9) | 47,274 (16.3) | NA | NA | NA | NA | 594 (27.8) | 5,382 (7.9) | 0 (0) | 10 (0) |
| | Pacific | 54,245 (5.9) | 109 (5.0) | 2,304 (6.8) | 483 (3.4) | 12,734 (4.4) | 2,170 (100) | 36,314 (100) | NA | NA | 11 (0.5) | 108 (0.2) | 0 (0) | 12 (0) |
| | Unknown | 12,392 (1.3) | NA | NA | NA | NA | NA | NA | NA | NA | 4 (0.2) | 91 (0.1) | 227 (1.8) | 12,070 (3.2) |

(*Continued*)

**Table 1.** (Continued)

| Partners | | Total[12] | A | | B | | C | | D | | E | | F | |
|---|---|---|---|---|---|---|---|---|---|---|---|---|---|---|
| | | | N = 36,319 (%) | | N = 303,214 (%) | | N = 38,484 (%) | | N = 85,034 (%) | | N = 70,313 (%) | | N = 393,653 (%) | |
| Presenting Symptoms at the time of RNA test[1,5] | Fever >100.4 F | 48,215 (5.2) | 440 (20.1) | 6,573 (19.3) | 2,656 (18.9) | 38,546 (13.3) | NA | NA | NA | NA | NA | NA | NA | NA |
| | Diarrhea | 12,394 (1.3) | NA | NA | 453 (3.2) | 6,910 (2.4) | 75 (3.4) | 1,106 (3.0) | 146 (5.2) | 3,704 (4.5) | NA | NA | NA | NA |
| | Chest pain | 17,287 (1.9) | 117 (5.3) | 1,772 (5.2) | 706 (5.1) | 9,394 (3.2) | 218 (10.0) | 1,867 (5.1) | 108 (3.8) | 3,105 (3.8) | NA | NA | NA | NA |
| | Delirium /Confusion | 6,474 (0.7) | 63 (2.9) | 1,734 (5.1) | 88 (0.6) | 2,165 (0.7) | 14 (0.6) | 187 (0.5) | 181 (6.4) | 2,042 (2.5) | NA | NA | NA | NA |
| | Headache | 17,416 (1.9) | 95 (4.3) | 1,630 (4.8) | 527 (3.7) | 6,449 (2.2) | 39 (1.8) | 995 (2.7) | 133 (4.7) | 7,548 (9.2) | NA | NA | NA | NA |
| | Sore throat | 23,551 (2.5) | 83 (3.8) | 1,411 (4.1) | 748 (5.3) | 14,656 (5.1) | NA | NA | 93 (3.3) | 6,560 (8.0) | NA | NA | NA | NA |
| | Cough | 98,111 (10.6) | 634 (28.9) | 8,644 (25.3) | 4,094 (29.1) | 59,693 (20.6) | 190 (8.7) | 4,810 (13.2) | 464 (16.5) | 19,582 (23.8) | NA | NA | NA | NA |
| | Shortness of breath | 51,526 (5.6) | 329 (15.0) | 5,765 (16.9) | 1,956 (13.9) | 26,374 (9.1) | 336 (15.4) | 3,623 (10.0) | 568 (20.2) | 12,575 (15.3) | NA | NA | NA | NA |
| | Pneumonia | 45,195 (4.9) | 268 (12.2) | 4,967 (14.6) | 1,462 (10.4) | 20,092 (6.9) | 324 (14.9) | 3,536 (9.7) | 1,049 (37.4) | 13,497 (16.4) | NA | NA | NA | NA |
| | Acute respiratory infection | 57,898 (0.7) | 255 (11.6) | 3,577 (10.5) | 2,194 (15.6) | 35,282 (12.2) | 56 (2.6) | 1,718 (4.7) | 867 (30.9) | 13,949 (17.0) | NA | NA | NA | NA |
| | Acute respiratory distress, arrest, or failure | 6,819 (0.7) | 110 (5.0) | 2,435 (7.1) | NA | NA | 282 (13.0) | 2,867 (7.9) | 32 (1.1) | 1,093 (1.3) | NA | NA | NA | NA |
| | Acute bronchitis | 1,837 (0.2) | 15 (0.7) | 191 (0.6) | 114 (0.8) | 1,206 (0.4) | NA | NA | 17 (0.6) | 294 (0.4) | NA | NA | NA | NA |
| | Sepsis | 9,038 (1.0) | 186 (8.5) | 4,067 (11.9) | NA | NA | NA | NA | 433 (15.4) | 4,352 (5.3) | NA | NA | NA | NA |
| | Cardiovascular condition | 69,730 (7.5) | 397 (18.1) | 7,477 (21.9) | 2,546 (18.1) | 37,381 (12.9) | 624 (28.7) | 6,149 (16.9) | 981 (34.9) | 14,175 (17.2) | NA | NA | NA | NA |
| | Renal Condition | 14,569 (1.6) | 64 (2.9) | 2,412 (7.1) | 487 (3.5) | 9,117 (3.1) | 257 (11.8) | 2,232 (6.1) | NA | NA | NA | NA | NA | NA |
| Care Setting (where RNA test occurred)[9] | Outpatient | 368,217 (39.0) | 835 (38.1) | 12,370 (36.3) | 13,365 (95.1) | 278,125 (96.2) | NA | NA | NA | NA | 1,344 (62.9) | 62,178 (91.2) | NA | NA |
| | Inpatient | 7,744 (0.8) | 62 (2.8) | 1,567 (4.6) | 258 (1.8) | 4,456 (1.5) | NA | NA | NA | NA | 281 (13.2) | 1,120 (1.6) | NA | NA |
| | Emergency department | 8,840 (1.0) | 93 (4.2) | 2,211 (6.5) | 67 (0.5) | 1,079 (0.4) | NA | NA | NA | NA | 512 (24.0) | 4,878 (7.2) | NA | NA |
| | Urgent Care | 2,369 (0.3) | 203 (9.3) | 2,166 (6.3) | NA | NA | NA | NA | NA | NA | 0 (0.0) | 0 (0.0) | NA | NA |
| | Other | 5,864 (0.6) | NA | NA | 369 (2.6) | 5,495 (1.9) | NA | NA | NA | NA | 0 (0.0) | 0 (0.0) | NA | NA |

(*Continued*)

**Table 1.** (Continued)

| Partners | | Total[12] | A | | B | | C | | D | | E | | F | |
|---|---|---|---|---|---|---|---|---|---|---|---|---|---|---|
| | | | **N = 36,319 (%)** | | **N = 303,214 (%)** | | **N = 38,484 (%)** | | **N = 85,034 (%)** | | **N = 70,313 (%)** | | **N = 393,653 (%)** | |
| Calendar Time[10] (based on RNA test) | Before May 1, 2020 | 67,093 (7.2) | 1,401 (63.9) | 15,148 (44.4) | 2,976 (21.2) | 17,933 (6.2) | 364 (16.8) | 3,115 (8.5) | 238 (8.5) | 9,807 (11.9) | 258 (12.1) | 784 (1.2) | 707 (5.7) | 14,362 (3.8) |
| | On/After May 1, 2020 | 859,924 (92.8) | 790 (36.1) | 18,980 (55.6) | 11,083 (78.8) | 271,222 (93.8) | 1,806 (83.2) | 33,199 (91.4) | 2,570 (91.5) | 72,419 (88.1) | 1,879 (87.9) | 67,392 (98.8) | 11,734 (94.3) | 366,850 (96.2) |

[1] At the time of RNA or serological sample refers to +/- 14 days from the sample collection date for the relevant test

[2] The unaccounted samples in Partners A and B were missing.

[3] Data was not available

[4] Hispanic ethnicity was not mutually exclusive from race.

[5] Phenotypes (code-sets) of ICD-10, medication, and LOINC are provided in S2 Table. Conditions may be identified using ICD-10, medication, or both.

[6] Pre-existing conditions were assessed 365 days before the index date and were not mutually exclusive.

[7] Pregnancy Status was assessed up to 40 weeks before the index date.

[8] Geographic regions were based on patients' home zip codes and defined by the US Census Bureau (https://www2.census.gov/geo/pdfs/maps-data/maps/reference/us_regdiv.pdf) and mapped by census track zip code. States included in each region are as follows: **New England**: Connecticut, Maine, Massachusetts, New Hampshire, Rhode Island, Vermont; **Mid Atlantic**: New Jersey, New York, Pennsylvania; **East North Central**: Indiana, Illinois, Michigan, Ohio, Wisconsin; **West North Central**: Iowa, Nebraska, Kansas, North Dakota, Minnesota, South Dakota, Missouri; **South Atlantic**: Delaware, District of Columbia, Florida, Georgia, Maryland, North Carolina, South Carolina, Virginia, West Virginia; **East South Central**: Alabama, Kentucky, Mississippi, Tennessee; **West South Central**: Arkansas, Louisiana, Oklahoma, Texas; **Mountain:** Arizona, Colorado, Idaho, New Mexico, Montana, Utah, Nevada, Wyoming; **Pacific:** Alaska, California, Hawaii, Oregon, Washington.

[9] The unaccounted samples in Partner A were missing.

[10] The FDA issued guidance for clinical laboratories, commercial manufacturers, and FDA staff on the use of diagnostic and serological tests for COVID-19 on May 16, 2020. https://www.fda.gov/news-events/fda-voices/insight-fdas-revised-policy-antibody-tests-prioritizing-access-and-accuracy.

[11] No pre-existing conditions—defined as those identified to have none of the above listed preexisting conditions.

[12] Because some partners did not collect and report some variables, care should be taken when interpreting the total number of each variable.

Serology manufacturer and test name were captured by four analytic partners, and mostly complete (<30% missing) for three included in this analysis (A, C, E). One of our largest partners was missing manufacturer data in 85% of the sample, and two partners were missing it completely. While manufacturer and assay name, as well as other metadata, are typically captured and available for export from the instrument, oftentimes laboratory information systems are not configured to receive or store this information. Constraints on integration include technical limitations of software and middleware, as well as a lack of clinical need, business case, or regulatory incentive. Capturing, storing, and transferring this additional data would require a substantial investment of resources to modify and/or reconfigure existing instruments, laboratory information systems, connective middleware, and EHRs. Absent a regulatory or reimbursement requirement, companies perceive little need to invest such resources given other competing priorities.

## Serotesting by demographic characteristics

Overall, we observed a higher distribution of persons aged 45–64 among those serotested compared to those not serotested. Four partners representing healthcare delivery systems reported race with <30% missing. Across three of these partners, we observed a higher distribution of White individuals among those serotested compared to those not. We did not observe a consistent pattern in serotesting by sex.

Five partners had representation across more than one region of the U.S. In partners with national representation, patients from the West North Central (Iowa, Nebraska, Kansas, North Dakota, Minnesota, South Dakota, Missouri) and West South Central (Arkansas, Louisiana,

**Table 2. Characterization of molecular and serologic tests among those with follow up serology test.**

| Partners | | A | B | C | D | E | F |
|---|---|---|---|---|---|---|---|
| | | N = 2,191 (%) | N = 14,059(%) | N = 2,170(%) | N = 2,808(%) | N = 2,137(%) | N = 12,441 (%) |
| Serological Test Type[1] | IgG | 2,073 (94.6) | 12,480 (88.8) | 2,170 (100) | 2,738 (97.5) | 849 (39.7) | 9,916 (79.7) |
| | Total Antibody | 105 (4.8) | 1,744 (12.4) | NA[2] | 41 (1.5) | 1288 (60.3) | 2,044 (16.4) |
| | Combined | 13 (0.06) | NA | NA | 29 (1.0) | NA | 481 (3.9) |
| Molecular Test Type | NAAT[3] | NA | NA | NA | NA | NA | 393,653 (100) |
| | N gene | NA | 6 (0.1) | NA | NA | NA | NA |
| | RNA | NA | 13,979 (99.4) | 36,314 (100) | NA | NA | NA |
| | RdRp gene[4] | NA | 26 (0.2) | NA | NA | NA | NA |
| | Other | NA | 48 (0.3) | NA | NA | NA | NA |
| Manufacturer—serological test name[5] | Γ | 371 (16.9) | 668 (4.7) | NA | NA | 496 (23.2) | NA |
| | Δ | 1,318 (60.2) | 1,423 (10.1) | NA | NA | NA | NA |
| | Θ | NA | NA | 531 (24.47) | NA | NA | NA |
| | Λ | NA | NA | 941 (43.36) | NA | NA | NA |
| | Ξ | NA | NA | 207 (9.5) | NA | NA | NA |
| | Π | 1 (0.04) | NA | NA | NA | 353 (16.5) | NA |
| | Ψ | NA | NA | NA | NA | 1,288 (60.3) | NA |
| | Unknown/Missing | 501 (22.9) | 11,968 (85.1) | 491 (22.6) | NA | NA[6] | NA |
| Manufacturer—molecular test name[5] | Σ | 210 (9.6) | 541 (3.8) | NA | NA | NA | NA |
| | Φ | NA | 126 (0.9) | NA | NA | NA | NA |
| | Ω | 41 (1.9) | NA | NA | NA | NA | NA |
| | X | NA | 597 (4.2) | NA | NA | NA | NA |
| | Y | 150 (6.8) | 83 (0.6) | NA | NA | NA | NA |
| | Unknown/Missing | 1,790 (81.7) | 12712 (90.4) | NA | NA | NA | NA |
| Molecular test Sample Type | Respiratory | NA | 13,784 (98.1) | NA | NA | NA | 259,744 (66.0)[7] |
| | Nasopharyngeal Swab | NA | 8 (0.1) | NA | NA | NA | NA |
| | Unknown/Missing | NA | 267 (1.9) | NA | NA | NA | 133,909 (34.0)[7] |

[1] The sum for Partners B exceeded the total sample because 165 patients, respectively, received a test for both IgG and Total Antibody and were counted in both groups.

[2] Data was not available

[3] Nucleic Acid Amplification Test

[4] RNA-dependent RNA polymerase gene

[5] We refer to the tests as Γ—Y for the purposes of anonymity. Most tests received an Emergency Use Authorization from the FDA. References available upon request

[6] The sum for Partner E's manufacturer-serological test name is classified as unknown/missing.

[7] The sum between the molecular test sample type for Partner F includes all people that have a positive RNA test result.

Oklahoma, Texas) regions were under-represented among the serotested. Two partners operated primarily in a single U.S. state and thus did not allow assessment of geographic differences.

## Serotesting by care-setting, symptoms, and pre-existing conditions

Half of the partners reported care-setting. Generally, most of the population was seen in the outpatient setting for their index visit. Large national insurer data did not suggest any differences in the distribution of index visit care settings among serotested vs. non-serotested. However, EHR data from a large national health data consortium revealed a higher distribution of patients in the inpatient setting among the serotested compared to non-serotested (13% vs. 2%, respectively).

As shown in Table 3, four of six partners reported presenting symptoms at index. Patterns in serotesting by symptoms seem to align with the data source. In partners who relied on

**Table 3. Clinical presentation and concomitant influenza or other viral infection around the time of serological sampling.**

| Partners | | A | B | C | D | E | F |
|---|---|---|---|---|---|---|---|
| | | N = 2,191 (%) | N = 14,059 (%) | N = 2,170 (%) | N = 2,808 (%) | N = 2,137 (%) | N = 12,441 (%) |
| Symptoms around the time of Serology test[2,3] | Any Chargemaster[1] or Medical Claim | 1,743 (79.6) | 14,059 (100) | NA[4] | NA | NA | NA |
| | Fever >100.4 F | 86 (3.9) | 675 (4.8) | NA | NA | NA | NA |
| | Diarrhea | NA | 188 (1.3) | 75 (3.4) | 101 (3.6) | NA | NA |
| | Chest pain | 91 (4.2) | 579 (4.1) | 218 (10.1) | 108 (3.8) | NA | NA |
| | Delirium/Confusion | 39 (1.8) | 76 (0.5) | 14 (0.6) | 279 (9.9) | NA | NA |
| | Headache | 39 (1.8) | 257 (1.8) | 39 (1.8) | 66 (2.3) | NA | NA |
| | Sore throat | 41 (1.9) | 291 (2.1) | NA | 31 (1.1) | NA | NA |
| | Cough | 199 (9.1) | 1,657 (11.8) | 190 (8.7) | 194 (6.9) | NA | NA |
| | Shortness of breath | 152 (6.9) | 1,072 (7.6) | 336 (15.4) | 430 (15.3) | NA | NA |
| | Pneumonia | 118 (5.4) | 899 (6.4) | 324 (14.9) | 1,046 (37.2) | NA | NA |
| | Acute Bronchitis | 9 (0.4) | 46 (0.3) | NA | 10 (0.4) | NA | NA |
| | Acute respiratory infection | 55 (2.5) | 781 (5.6) | 56 (2.6) | 927 (33.0) | NA | NA |
| | Acute respiratory distress, arrest | 42 (1.9) | NA | 282 (13.0) | 24 (0.8) | NA | NA |
| | Cardiovascular condition | 309 (14.1) | 3,047 (21.7) | 624 (28.7) | 1,159 (41.3) | NA | NA |
| | Renal Condition | 38 (1.7) | 437 (3.1) | 257 (11.8) | NA | NA | NA |
| | Known exposure to COVID-19 | 724 (33.0) | 6,668 (47.4) | NA | 406 (14.5) | NA | NA |

[1] A hospital chargemaster is a comprehensive list of a hospital's products, procedures, and services that could produce a charge. It will have a record for everything in the health system that relates to patient care.

[2] At the time of RNA or serological sampling refers to +/- 14 days the from sample collection date. Symptoms are not mutually exclusive.

[3] Phenotypes (code-sets) of ICD-10, medication, and LOINC are provided in S2 Table. Conditions may be identified using ICD-10.

[4] Data was not available

claims data, we generally see no systematic trend in serotesting by presenting symptoms at the time of the index visit. Among systems that relied on EHR data, we see a higher distribution of patients with shortness of breath (15–20%), pneumonia (15–37%), and cardiovascular conditions (29–35%) among the serotested vs. non-serotested (10–15%, 10–16%, 17%, respectively).

All but one data partner reported pre-existing conditions. We found individuals with pre-existing cardiovascular disease tended to have greater representation in the serotested (35%–48%) vs. non-serotested group (17%–40%). In partners with EHR data, a greater distribution of patients with pre-existing obesity and kidney disease were also observed among the serotested compared to non-serotested. We did not observe a differential in testing among pregnant women–although only half of the contributing partners reported pregnancy status. We observed similar patterns of pregnancy among women with serological testing (4–13%) compared with women without serological testing (2–11%), with a slightly higher range in prevalence of pregnancy among women with serological testing.

As shown in Table 3 and Fig 3, many of the same symptoms at the time of RNA testing persisted at the time of serotesting, which may be attributed to the high volume of same-day molecular and serological testing.

## Frequency and time to serological testing

In all but one healthcare system, serological testing increased substantially after May 1, 2020 (Table 1). Serological testing among those with positive RNA ranged from 3–6% across our

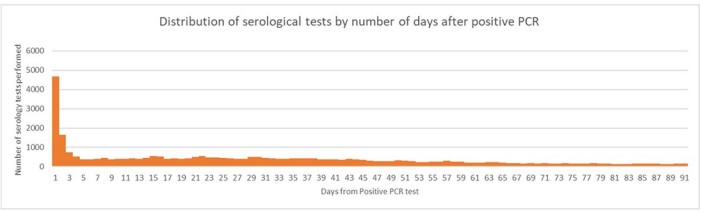

**Fig 3. Distribution of serological tests by the number of days after positive RNA test.**

contributing partners. Among all people with follow-up samples, 15% had a follow-up serology within 14 days of the index RNA test (Fig 3).

Overall, the median time to serotesting from RNA per data partner ranged from 10–31 days and was shorter in datasets from systems with data from EHRs (Table 4). In terms of age, adults 85 years and older tended to have the shortest time to follow-up between molecular and serology testing (median range: 1–25 days). In partners with robust capture of race and ethnicity, Black patients (median: 7–15 days) tended to experience a shorter time to serotesting as compared to White individuals (median: 13–21 days). In half of the analytic datasets, time to serotesting tended to be shortest in people with a history of dementia (median: 2–15 days). Within and across datasets, there was substantial variability in time to serotesting by presenting symptoms at index. In the two partners reporting on pregnancy, time to serotesting did not tend to differ by pregnancy status.

In general, we did not observe repeat molecular or serological testing within the 90–day time frame. In partners A–E, the median (IQR) number of both tests was 1 (0); while in partner F it was 1 (1). Time to serotesting tended to be shorter for IgG tests as compared to total antibody. There was substantial variation in time to serological testing across manufacturer assays (both molecular and serological). We observed differences in time to serological testing across care settings in only one dataset, with the median time to serotesting being 0 in the inpatient setting and almost one month in the outpatient. Patients with index dates after May 1st, 2020 tended to wait fewer days for serological testing (median: 7–27) compared to those with index before May 1st, 2020 (median: 28–43). This difference may be explained by the lower availability of SARS-CoV-2 tests before May 1 since serology tests were not authorized before April 15, 2020; and molecular tests were not authorized before March 15, 2020.

## Discussion

The Centers for Disease Control has initiated several large-scale population-based seroprevalence studies throughout the U.S. [24]. We conducted this study to characterize the real-world use of COVID-19 serological testing. We identified a number of key findings: 1) a substantial proportion of serology tests were conducted within 14 days of the RNA test, the majority of which occurred on the same day as the positive RNA test; 2) a lack of data interoperability between the instrument, laboratory, and clinical data could limit the ability to conduct a large-scale assessment of the real-world performance of not only COVID-19 tests, but other diagnostic and laboratory tests; 3) missing race/ethnicity data may impede a comprehensive understanding of racial disparities involved in COVID-19 serology and immunity, and 4) important differences in the testing landscape presented from claims vs. EHR data sources may impact results generated from these data sources.

We assumed the date of a positive SARS-CoV-2 molecular test would be a reasonable proxy for symptom onset. We did not expect that 15% of serotesting would occur within 14

**Table 4. Characterization of the timing of serology testing relative to RNA sampling date.**

| Partners | | A | B | C | D | E | F |
|---|---|---|---|---|---|---|---|
| | | Median days to serology test (from positive RNA test) (25–75 percentile) | | | | | |
| Age at the time of RNA test (Years)[2,3] | 0–3 | 32 (32, 32) | 36 (24, 43) | 8 (1,15) | NA[1] | 0 (0,3) | 3 (0, 26) |
| | 4–11 | 27 (15, 33) | 29 (17, 43) | 2 (1,8) | 19 (4,26) | 27 (1,38) | 12 (1, 39) |
| | 12–17 | 18 (5, 36) | 28 (12, 53) | 1 (1,13) | 17 (2,34) | 34 (2,55) | 28 (1,54) |
| | 18–44 | 29 (11, 45) | 28 (12, 49) | 14 (1,36) | 14 (0,41) | 27 (2,40) | 31 (11, 55) |
| | 45–54 | 32 (15, 45) | 32 (16, 52) | 13 (1,38) | 13 (1,39) | 27 (1,41) | 30 (10, 54) |
| | 55–64 | 35 (21, 51) | 35 (19, 53) | 13 (1,34) | 14 (1,40) | 21 (1,38) | 28 (8, 51) |
| | 65–74 | 32 (15, 45) | 33 (19, 51) | 8 (1,29) | 13 (1,36) | 10 (1,32) | 19 (3, 44) |
| | 75–84 | 32 (11, 53) | 32 (15, 52) | 6 (1,22) | 8 (1,28) | 6 (0,3) | 10 (2, 30) |
| | 85+ | 21 (1, 46) | 25 (9, 49) | 1 (1,14) | 7 (1,27) | 2 (0,16) | 6 (1, 18) |
| | Overall[3] | 31 (15, 46) | 31 (15, 51) | 10 (1,34) | 12 (1,38) | 20 (1,37) | 24 (5, 49) |
| Sex | Female | 31 (15, 47) | 33 (16, 52) | 14 (1,37) | 13 (1,43) | 25 (1,41) | 28 (7, 52) |
| | Male | 31 (14, 45) | 29 (14, 49) | 6 (1,29) | 10 (1,32) | 13 (0,33) | 20 (4, 45) |
| | Unknown/ Missing | 28 (7, 48) | 35 (20, 49) | NA | NA | NA | 27 (23,32) |
| Race/Ethnicity (not mutually exclusive) | White | NA | 33 (16, 54) | 13 (1,35) | 12 (1,39) | 21 (1,38) | 26 (7, 51) |
| | Black | NA | 34 (17, 57) | 7 (1,30) | 12 (1,40) | 11 (0,31) | 12 (1, 36) |
| | Asian | NA | 38 (28, 55) | 7 (1,28) | 15 (2,32) | 20 (1,33) | 16 (4, 37) |
| | Pacific Islander/Native Hawaiian | NA | NA | 10 (1,44) | 33 (10,57) | 22 (0,38) | NA |
| | Hispanic or Latino | NA | 29 (15, 49) | 5 (1,30) | 5 (1,27) | 21 (2,38) | 11 (1, 33) |
| | American Indian or Alaska Native | NA | 20 (16, 25) | 10 (6,17) | 3.5 (0,19) | 0 (0,14) | 36 (11, 45) |
| | Other | NA | NA | 2 (1,14) | 38 (1,39) | NA | 17 (1, 41) |
| | Unknown | NA | 31 (14, 50) | 13 (1,35) | 21 (7,37) | 22 (1,40) | NA |

(*Continued*)

**Table 4.** (Continued)

| Partners | | A | B | C | D | E | F |
|---|---|---|---|---|---|---|---|
| | | **Median days to serology test (from positive RNA test) (25–75 percentile)** | | | | | |
| **Pre-existing Conditions**[4,5] | Cardiovascular disease | 33 (18, 49) | 33 (17, 52) | 8 (1,31) | 40 (2,42) | NA | 10 (2, 32) |
| | Diabetes | 33 (16, 49) | 32 (16, 51) | 5 (1,26) | 39 (1,40) | NA | 10 (2, 33) |
| | Hypertension | 33 (17, 50) | 33 (17, 52) | 9 (1,34) | NA | NA | 11 (2, 35) |
| | Cancer | 35 (15, 52) | NA | 7 (1,27) | 36 (1,37) | NA | 15 (3, 41) |
| | Asthma | 30 (13, 44) | 35 (18, 54) | 11 (1,36) | 46 (3,49) | NA | 11 (1, 26) |
| | Kidney Disease | 29 (5, 48) | 32 (15, 51) | 6 (1,33) | 37 (2,39) | NA | 8 (1, 27) |
| | Chronic Lung conditions | 32 (15, 46) | 33 (17, 51) | NA | 42 (3,45) | NA | 7 (6, 49) |
| | Auto-Immune conditions | 33 (14, 51) | 34 (18, 53) | 10 (1,36) | NA | NA | 25 (1, 41) |
| | HIV | 40 (30, 66) | NA | 8 (1,39) | 50 (7,57) | NA | 18 (2, 33) |
| | Any liver disease | 29 (14, 48) | 34 (18, 53) | 4 (1,24) | 37 (0,37) | NA | 11 (4, 37) |
| | Obesity | 33 (16, 48) | 30 (15, 50) | 4 (1,33) | 47 (2,49) | NA | 19 (3, 45) |
| | Dementia | 23 (1, 41) | NA | 2 (1,17) | 13 (1,14) | NA | 7 (2, 21) |
| **Pregnancy Status**[6] | No | NA | NA | 10 (1,34) | 37 (1,38) | NA | NA |
| | Yes | NA | NA | 10 (1,29) | 48 (0,48) | NA | NA |
| **Geography (patient residence)**[7] | Mid-Atlantic | 33 (17, 48) | 35 (17, 53) | NA | NA | 3 (0,25) | NA |
| | New England | 38 (12, 57) | 35 (19, 53) | NA | NA | 12 (12,12) | NA |
| | South-Atlantic | 29 (18, 44) | 32 (17, 54) | NA | NA | 2 (0,15) | NA |
| | East North Central | 25 (8, 38) | 31 (15, 53) | NA | NA | 7 (0,33) | 25 (5,49) |
| | East South Central | 37 (19, 60) | 28 (12, 48) | NA | NA | 1 (0,4) | 28 (4,45) |
| | West North Central | 29 (17, 49) | 34 (18, 54) | NA | NA | 22 (1,41) | NA |
| | West South Central | 25 (8, 38) | 23 (9, 42) | NA | NA | 7 (0,18) | 16 (9,24) |
| | Mountain | 34 (15, 46) | 29 (14, 47) | NA | NA | 30 (11,40) | NA |
| | Pacific | 29 (9,50) | 30 (13, 47) | 10 (1,34) | NA | 22 (2,29) | NA |
| | Unknown | NA | 24 (14, 40) | NA | NA | 0 (0,4) | 2 (1,19) |

(*Continued*)

**Table 4.** (Continued)

| Partners | | A | B | C | D | E | F |
|---|---|---|---|---|---|---|---|
| | | Median days to serology test (from positive RNA test) (25–75 percentile) | | | | | |
| **Presenting Symptoms at the time of RNA test**[2,3] | Fever >100.4 F | 37 (24, 49) | 37 (22, 55) | NA | NA | NA | NA |
| | Diarrhea | NA | 34 (19, 53) | 6 (1,33) | 39 (3,42) | NA | NA |
| | Hypoglycemic | NA | NA | NA | NA | NA | NA |
| | Chest pain | 36 (22, 50) | 31 (16, 48) | 2 (1,14) | 34 (4,38) | NA | NA |
| | Delirium/Confusion | 39 (13, 55) | 17 (2, 39) | 1 (1,5) | 11 (1,12) | NA | NA |
| | Headache | 38 (22, 59) | 33 (16, 54) | 2 (1,26) | 45 (7,52) | NA | NA |
| | Sore throat | 33 (12, 45) | 30 (15, 50) | NA | 39 (14,53) | NA | NA |
| | Cough | 37 (23, 49) | 35 (20, 53) | 24 (2,48) | 48 (8,56) | NA | NA |
| | Shortness of breath | 38 (24, 51) | 35 (19, 53) | 2 (1,16) | 34 (1,35) | NA | NA |
| | Pneumonia | 38 (25, 52) | 33 (16, 50) | 2 (1,17) | 22 (1,23) | NA | NA |
| | Acute bronchitis | 20 (7, 55) | 35 (23, 51) | NA | 42 (2,44) | NA | NA |
| | Acute respiratory infection | 38 (25, 51) | 36 (19, 54) | 22 (2,45) | 18 (1,19) | NA | NA |
| | Acute respiratory distress, arrest, or failure | 36 (22, 52) | NA | 2 (1,8) | 35 (1,36) | NA | NA |
| | Cardiovascular condition | 33 (18, 49) | 30 (14, 50) | 3 (1,19) | 30 (1,31) | NA | NA |
| | Renal Condition | NA | 29 (9, 47) | 3 (1,27) | NA | NA | NA |
| **Serological Test Type** | Total Antibody | 51 (13, 75) | 36 (19, 57) | NA | 37 (5,42) | 11 (0,36) | 36 (19, 58) |
| | IgG | 31 (15, 45) | 30 (15, 50) | 10 (1,34) | 49 (28,77) | 28 (7,38) | 38 (21, 60) |
| **Manufacturer–serological test name (assay)**[8] | Γ | 1 (1,1) | 33 (17, 48) | NA | NA | 22 (1,37) | NA |
| | Δ | 1 (1,1) | 26 (7, 45) | NA | NA | NA | NA |
| | Π | NA | NA | NA | NA | 32 (21,39) | NA |
| | Ξ | NA | NA | 4 (1,21) | NA | 11 (0,36) | NA |
| | Θ | NA | NA | 13 (1,17) | NA | NA | NA |
| | Ψ | NA | NA | NA | NA | 11 (0,36) | NA |
| | Λ | NA | NA | 23 (2,40) | NA | NA | NA |
| | Missing/Unknown | NA | NA | NA | NA | NA | NA |
| **Manufacturer–molecular test name (assay)**[8] | Ω | 15 (1, 30) | NA | NA | NA | NA | NA |
| | Υ | 1 (1, 21) | 15 (0, 35) | NA | NA | NA | NA |
| | Χ | NA | 21 (7, 41) | NA | NA | NA | NA |
| | Σ | 1 (1, 23) | 23 (9, 43) | NA | NA | NA | NA |
| | Φ | NA | 25 (12, 48) | NA | NA | NA | NA |
| | Missing/Unknown | NA | NA | NA | NA | NA | NA |

(*Continued*)

**Table 4.** (Continued)

| Partners | | A | B | C | D | E | F |
|---|---|---|---|---|---|---|---|
| | | **Median days to serology test (from positive RNA test) (25–75 percentile)** | | | | | |
| **Care Setting where RNA test occurred** | Inpatient | 37 (24,57) | 31 (6, 48) | 10 (1,12) | NA | 0 (0,3) | NA |
| | Outpatient | 35 (21, 50) | 31 (15, 51) | NA | NA | 30 (12,43) | NA |
| | Emergency department | 39 (31, 52) | 33 (21, 53) | NA | NA | 1 (0,21) | NA |
| **Calendar Time (based on RNA test)** | On or after May 1, 2020 | 9 (1, 27) | 27 (12, 48) | 71,29) | 32 (1,33) | 15 (0,35) | 22 (4, 47) |
| | Before May 1, 2020 | 39 (28, 51) | 43 (30, 58) | 28 (6, 50) | 38 (30,68) | 36 (25,49) | 46 (30, 64) |

[1] Data was not available

[2] At the time of RNA or serological sample refers to +/- 14 days from the sample collection date for the relevant test

[3] The median time to event across all participants

[4] Pre-existing conditions were assessed 365 before the index date.

[5] Phenotypes (code-sets) of ICD-10, medication, and LOINC are provided in S2 Table. Conditions may be identified using ICD-10, medication, or both.

[6] Pregnancy Status was assessed up to 40 weeks before the index date.

[7] The geographic regions were based on the regions defined by the US Census Bureau and are taken from https://www2.census.gov/geo/pdfs/maps-data/maps/reference/us_regdiv.pdf. The states that are included in each region are New England: Connecticut, Maine, Massachusetts, New Hampshire, Rhode Island, Vermont; Mid Atlantic: New Jersey, New York, Pennsylvania; East North Central: Indiana, Illinois, Michigan, Ohio, Wisconsin; West North Central: Iowa, Nebraska, Kansas, North Dakota, Minnesota, South Dakota, Missouri; South Atlantic: Delaware, District of Columbia, Florida, Georgia, Maryland, North Carolina, South Carolina, Virginia, West Virginia; East South Central: Alabama, Kentucky, Mississippi, Tennessee; West South Central: Arkansas, Louisiana, Oklahoma, Texas; Mountain: Arizona, Colorado, Idaho, New Mexico, Montana, Utah, Nevada, Wyoming; Pacific: Alaska, California, Hawaii, Oregon, Washington

[8] We refer to the tests as Γ—Υ for the purposes of anonymity. Some of the tests received an emergency use authorization (EUA). References available upon request

days of the RNA test, and most often on the same day. This is an important finding because we would not expect concordance between molecular and serology tests taken in close proximity because of known viral kinetics [25–27] After consulting with our analytic partners, we discovered the implementation of policies within health systems to screen patients admitted for procedures for active or past SARS-CoV-2 to evaluate the risk of nosocomial infections. These policies may be driving observed differences in the median time between molecular and serology tests in claims (31 days), compared to EHR datasets (10–24 days), with the nuance being washed out in larger claims datasets that incorporate a mix of care settings. Clinicians may also be serotesting because they do not believe that patients are presenting close to the time of exposure, desire a better understanding of patients' disease progression, or to assist in determining clinical course of care, which may depend on whether patients are at increased risk for severe illness due to insufficient antibody response [28]. For all diagnostic and serological tests authorized by the FDA, the FDA produces fact sheets for healthcare providers to provide information about the assay and its limitations [29]. Continued guidance and communication are needed to help clinicians understand how to best use serological tests for SARS-CoV-2 [30, 31].

A higher distribution of patients presenting with respiratory, metabolic, and cardiovascular symptoms among the serotested compared to non-serotested, is consistent with an evaluation by the CDC that indicated such factors are associated with severe COVID-19 illness [32]. Patients with a pre-existing history of cardiovascular disease (including hypertension) and liver disease were over-represented among those serotested vs. those not serotested in multiple datasets. These conditions have been shown to be associated with excess risk in other studies

[33, 34]. It was surprising that we did not observe any differences in the distribution of cancer in those serotested compared to the non-serotested. More research is needed to understand why some patients with known active SARS-CoV-2 infection receive a serology test, while others do not.

Across care delivery systems, a notable observation was increased serological testing in White as compared to Black individuals. However, when Black patients did receive serology testing, the time to testing was shorter, which may be due to a pressing need to identify the presence of antibodies/past infection in populations who have been shown to be at higher risk of COVID-19 morbidity and mortality [17]. More importantly, data on race from a large national insurer was missing in about 80% of the sample. Without data on race and ethnicity, the racial disparities in COVID-19 outcomes—and healthcare in general–cannot be addressed.

Another important information gap is in manufacturer data. Despite targeted conversations with technology teams and experts in technical, syntactic, and semantic interoperability, only half of analytic partners were able to integrate test manufacturer data with LIS and EHR data. A lack of data interoperability within healthcare is a historic problem [35]. Such interoperability is the foundation for public health surveillance, research, artificial intelligence, medical advances, and quality assurance in the context of EUA [36, 37]. Healthcare systems, manufacturers, and information technology vendors should move to fill information gaps to improve response to COVID-19 and future public health threats.

## Differences in results reported by claims vs. EHR-based systems

Analytic partners ran their analyses in parallel and aligned on a common analytic plan. We did not pool data, which allowed us to highlight, rather than control for differences across partners. Different patterns between EHR and claims systems were apparent in our analysis. In general, claims datasets showed no difference in serotesting by care setting or presenting symptoms, whereas EHR systems did. And while all datasets showed an elevated prevalence of pre-existing cardiovascular disease observed among those serotested (compared to the non-serotested), EHR datasets also showed a greater distribution of people with pre-existing obesity, kidney disease, and chronic lung conditions among the serotested. Because healthcare delivery systems generally have a limited ability to capture all clinical events for a given patient [38], sicker patients may be driving identification within certain health systems and pre-existing conditions may have been missed in patients who do not regularly attend the facility for care but were diverted to the facility [38]. Our data support this hypothesis on both points of increased illness among patients and lower identification of pre-existing conditions among patients identified from EHR compared to claims data sources. These differences may influence the interpretation of serology tests [38–42].

## Strengths and limitations

Our study has many strengths. This was a large assessment of serotesting across the U.S. in diverse datasets leveraging either EHR or claims data. We developed a protocol that incorporated the unique characteristics of each data source and provided a forum to transparently communicate and collaborate on study design and interpretation. We also established a platform to rapidly collect and analyze data from various systems to evaluate process improvement and identify important trends over time. Such a platform may be used to evaluate process improvement and comparisons within data systems.

Our study also has some important limitations. First, we were unable to assess the independence of samples across the healthcare partners directly. Three partners provide national coverage, and thus large sample sizes. The geographic distribution of their populations does not

suggest overlap. However, single health systems included in the same geographic region as the larger healthcare partners (specifically in the Pacific and Mountain regions) may be double counted. Second, smoking status, BMI, and race were largely missing in our analysis. These are important characteristics in assessing the impact of COVID-19 on the health of the population. Third, the sample collection date was not always available the and result date was used by some partners. As such, it is possible that samples collected on the same day may have different result dates if tests were run sequentially. Fourth, manufacturer information was largely missing from two of our largest datasets because instrument data either did not flow to the laboratory information system (LIS), or those results were not transmitted from the LIS to the EHR or payer database. However, we did not find differential missingness by age, sex, or geography among individuals with and without manufacturer data. Finally, lack of data on COVID-19 exposure and symptom onset limits our ability to make future inferences on appropriate pairs of molecular and serological tests to assess serological performance for past infection. We note that assumptions regarding the proximity of RNA testing to symptom onset may not be reliable over time. Testing for active infection has gone from severely limited at the start of the pandemic (March-April 2020) to widely available today. People may receive serial RNA testing without suspected exposure for purposes of employment or recreational gathering with friends and family.

As in all observational datasets, the completeness of our assessment is dependent on the capture of events in each of our healthcare data partners. Indeed, we observed that a greater proportion (35–65%) of patients identified in EHR data had no encounter in the year prior to index, compared to 11% among those identified from payer data. Coupled with our observation from EHRs that there seemed to be a greater number of pre-existing conditions for which there was preferential serotesting, these data provide additional evidence that patients identified through EHR data sources may tend to be sicker than those identified in claims. Furthermore, not knowing "care setting" for a large portion of tests could affect interpretation of the performance of serology testing as well, since the sensitivity of serology assays appears to be lower in mildly sick and/or asymptomatic cohorts.

## Conclusion

Our results inform the underlying context of serotesting during the first year of the COVID-19 pandemic and differences in serotesting trends observed from claims and EHR data sources–a critical first step to understanding the real-world accuracy of serological tests. The limited ability to link test manufacturer data with lab results and clinical data, and incomplete reporting of race/ethnicity data challenge the ability to assess real-world performance of SARS-CoV-2 tests in different populations and settings. These shortcomings challenge the overall U.S. response to current and future disease pandemics.

## Supporting information

**S1 Table. Characteristics of participating data sources and representative populations.**
(DOCX)

**S2 Table. Phenotype (code-lists) for specified presenting symptoms & pre-existing conditions.**
(DOCX)

**S1 Fig. Factors potentially associated with serological testing.**
(TIF)

## Acknowledgments

We would like to thank Christina Silcox, Shamiram Feinglass, Roland Romero, James Okusa, Elijah Mari Quinicot, Amar Bhat, Susan Winckler, Alecia Clary, Sadiqa Mahmood, Philip Ballentine, Perry L. Mar, Cynthia Lim Louis, Connor McAndrews, Elitza S. Theel, Cora Han, Pagan Morris, Charles Wilson, and Bridgit O Crews for their engagement, and assistance with this manuscript. We would also like to note Daniel Caños, Sara Brenner, Wendy Rubinstein, Veronica Sansing-Foster, and Sean Tunis for their support and feedback during this work. A special thanks and recognition for the contributions and sacrifice of Dr. Michael Waters, our dear colleague and friend who will be forever in our thoughts. We thank Amir Alishahi Tabriz MD, PhD for his assistance with manuscript preparation.

## Author Contributions

**Conceptualization:** Carla V. Rodriguez-Watson.

**Data curation:** Natalie E. Sheils, Anthony M. Louder, Elizabeth H. Eldridge, Nancy D. Lin, Benjamin D. Pollock, Jennifer L. Gatz, Shaun J. Grannis, Rohit Vashisht, Carly Kabelac, Camille Knepper, Sandy Leonard, Peter J. Embi, William G. Jenkinson, Reyna Klesh, Omai B. Garner, Ayan Patel, Lisa Dahm, Aiden Barin, Dan M. Cooper, Tom Andriola, Carrie L. Byington, Bridgit O. Crews.

**Formal analysis:** Carla V. Rodriguez-Watson, Natalie E. Sheils, Anthony M. Louder, Elizabeth H. Eldridge, Nancy D. Lin, Benjamin D. Pollock, Jennifer L. Gatz, Shaun J. Grannis, Rohit Vashisht, Carly Kabelac, Camille Knepper, Peter J. Embi, William G. Jenkinson, Reyna Klesh, Ayan Patel.

**Funding acquisition:** Carla V. Rodriguez-Watson, Atul J. Butte.

**Investigation:** Carla V. Rodriguez-Watson, Natalie E. Sheils, Anthony M. Louder, Elizabeth H. Eldridge, Nancy D. Lin, Benjamin D. Pollock, Jennifer L. Gatz, Shaun J. Grannis, Rohit Vashisht, Carly Kabelac, Camille Knepper, Sandy Leonard, William G. Jenkinson, Reyna Klesh, Omai B. Garner, Ayan Patel, Lisa Dahm, Aiden Barin, Dan M. Cooper, Tom Andriola, Carrie L. Byington, Bridgit O. Crews, Atul J. Butte, Jeff Allen.

**Methodology:** Carla V. Rodriguez-Watson, Natalie E. Sheils, Anthony M. Louder, Elizabeth H. Eldridge, Nancy D. Lin, Benjamin D. Pollock, Jennifer L. Gatz, Shaun J. Grannis, Rohit Vashisht, Kanwal Ghauri, Gina Valo, Aloka G. Chakravarty, Tamar Lasky, Mary Jung, Stephen L. Lovell, Sandy Leonard, Peter J. Embi, William G. Jenkinson, Reyna Klesh, Omai B. Garner, Ayan Patel, Lisa Dahm, Aiden Barin, Dan M. Cooper, Tom Andriola, Carrie L. Byington, Bridgit O. Crews, Atul J. Butte, Jeff Allen.

**Project administration:** Carla V. Rodriguez-Watson, Kanwal Ghauri.

**Supervision:** Carla V. Rodriguez-Watson, Peter J. Embi, Atul J. Butte.

**Validation:** Carla V. Rodriguez-Watson, Aloka G. Chakravarty.

**Visualization:** Carla V. Rodriguez-Watson.

**Writing – original draft:** Carla V. Rodriguez-Watson, Natalie E. Sheils, Kanwal Ghauri.

**Writing – review & editing:** Carla V. Rodriguez-Watson, Natalie E. Sheils, Anthony M. Louder, Elizabeth H. Eldridge, Nancy D. Lin, Benjamin D. Pollock, Jennifer L. Gatz, Shaun J. Grannis, Rohit Vashisht, Kanwal Ghauri, Gina Valo, Aloka G. Chakravarty, Tamar Lasky, Mary Jung, Stephen L. Lovell, Jacqueline M. Major, Carly Kabelac, Camille Knepper, Sandy Leonard, Peter J. Embi, William G. Jenkinson, Reyna Klesh, Omai B. Garner, Ayan

Patel, Lisa Dahm, Aiden Barin, Dan M. Cooper, Tom Andriola, Carrie L. Byington, Bridgit O. Crews, Atul J. Butte, Jeff Allen.

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
