## [Decision Letter · Decision Letter 0]

15 Nov 2021

PONE-D-21-30196Real-world utilization of SARS-CoV-2 serological testing in RNA positive patients across the United StatesPLOS ONE

Dear Dr. Rodriguez-Watson,

Thank you for submitting your manuscript to PLOS ONE. After careful consideration, we feel that it has merit but does not fully meet PLOS ONE’s publication criteria as it currently stands. Therefore, we invite you to submit a revised version of the manuscript that addresses the points raised during the review process.

We look forward to receiving your revised manuscript.

Kind regards,

Jue Liu, Ph.D.

Academic Editor

PLOS ONE

Journal Requirements:

2. We note that Figure 1 in your submission contain map images which may be copyrighted. All PLOS content is published under the Creative Commons Attribution License (CC BY 4.0), which means that the manuscript, images, and Supporting Information files will be freely available online, and any third party is permitted to access, download, copy, distribute, and use these materials in any way, even commercially, with proper attribution. For these reasons, we cannot publish previously copyrighted maps or satellite images created using proprietary data, such as Google software (Google Maps, Street View, and Earth). For more information, see our copyright guidelines: http://journals.plos.org/plosone/s/licenses-and-copyright.

Reviewers' comments:

Reviewer's Responses to Questions

**Comments to the Author**

1. Is the manuscript technically sound, and do the data support the conclusions?

Reviewer #1: Yes

Reviewer #2: Partly

Reviewer #3: Yes

2. Has the statistical analysis been performed appropriately and rigorously? 

Reviewer #1: Yes

Reviewer #2: Yes

Reviewer #3: Yes

3. Have the authors made all data underlying the findings in their manuscript fully available?

Reviewer #1: Yes

Reviewer #2: Yes

Reviewer #3: Yes

4. Is the manuscript presented in an intelligible fashion and written in standard English?

Reviewer #1: Yes

Reviewer #2: Yes

Reviewer #3: Yes

5. Review Comments to the Author

Reviewer #1: The authors conducted a review of electronic health records and/or claims data across six major health systems in the United States of America to assess the real-world utilisation of serological testing for SARS-CoV-2 in patients who had tested positive for SARS-CoV-2 via RNA previously. They identified 930,669 individuals and noted that approximately 15% of them underwent serological testing within 14-days of positive diagnosis via RNA.

Unfortunately the study was limited by its retrospective nature and its reliance on EHR and claims data and as such a fair amount of data is missing including race/ethnicity data, molecular and RNA test type and symptomatology. However, as the intention of this manuscript is to characterise the utilisation of serological testing for SARS-CoV-2 in a real world setting, this missing data does not compromise the integrity of the study but in fact highlights shortcomings in the health system, in terms of critical information capturing such as demographic and clinical data.

Furthermore, the manuscript also gives insights into health practices during the pandemic including non-adherence to clinical guidelines related to SARS-CoV-2 testing and the use of serological assay. This information is important as it can inform future practice and highlight the need to follow evidence based guidance during the pandemic.

Notably the data is heavily skewed towards Providers B and F (> 66% of the cohort) but as the data is presented individually this is clear. The manuscript may benefit from the inclusion of a "total" column in all tables although this may be difficult to express the data due to the large amount of missing information.

Thank you for submitting this manuscript for review.

Reviewer #2: This is an impressive dataset collected from multiple systems addressing principles of inter-operability, population coverage and linkage of laboratory results with detailed clinical metadata to address policy or public health needs. The main concern is whether the objectives set have made sufficient progress to justify publication at this point. It also would benefit from more detailed explanation of the research questions behind the objectives set (presumably by the FDA) to support presentation of the progress made in the form of a research publication, rather than alternatively as a published interim project progress report. It also seems aligned to the requirements of the US approach to FDA diagnostics approvals, implementation and guidance. This makes it a bit hard to clearly see the specific research questions as opposed to providing data to support decision making within the US diagnostics regulatory framework.

The comments below expand on these concerns which are raised from perspective of a non-US infectious diseases clinician involved in SARS-CoV-2 testing including serological tests. Consequently some concerns may be easily addressed or reflect misunderstanding due to lack of familiarity with the US setting and how this work fits in.

Abstract - conclusion: how exactly does this methodological approach actually address the question of determining performance data or linking with manufacturer and clinical data.

It does not seem appropriate to say the need for more efficient testing strategies without presenting the actual strategies and rationale for testing in these different settings at that time. Given the dynamic nature of this disease there is potential for findings of a study like this to not be relevant by the time a follow up study is done or recommendations are made for the future.

For example, in the study period serology testing was sometimes performed to help with diagnosis of what was a new condition. Antibodies appear around day 10 and so without vaccination or potential for prior infection, it was not inappropriate to perform serology to help with diagnosis. Duration of symptoms was an important clinical measure not collected in this study.

In addition, patients now get serology testing on admission to determine which SARS-CoV-2 RNA-PCT positive patients are antibody negative and might require Regeneron or similar

Finally, now that vaccination is widespread serology testing is used more to assess immunity (not prevalence/incidence of disease) or as a universal hospital admission test to inform infection control decisions (ie antibodies +/- history of vaccination = protected and less likely to acquire SARS-CoV-2).

Line 69 serology is not only done to determine prevalence estimates

Line 73 Please explain for those not familiar with US regulatory processes what the relevance of 510(k) is and what the link is between approvals and this collection of real world evidence

Line 81 -83. Interoperability and linkage of tests with metadata from EHR impacts on all of medicine but what are the advantages of this approach here for determining test performance and linking with clinical symptoms. Please explain the benefits compared with studies conducted at representative institutions / different clinical settings where local laboratory and clinical data can be collected alongside guidelines and rationale. Realise the national approach will get there but it is a long and complex road to get to the granularity required to meet stated objectives.

Line 93. The FDA asked for this consortium to be established with specific purpose. The consortium has a broad range of experts and links diverse providers. Line 100 sets 3 different objectives. How does the meeting of those objectives link into a research question and publication?

Line 109: How did these organisations came together and are representative of the organisations involved in serology. They presumably were not randomly selected. The authors do say heavy representation from certain states

Line 126. March to September was very early on in the pandemic. First wave. Diagnostics and guidelines have improved significantly. All tested parameters are very different now, hence making generalisability comments, even for the US, seems tenuous for today.

From line 128: it is hard to understand the locations and factors underpinning access to diagnostic tests from outside the US system to interpret results and assess potential biases and confounders

Line 251; Duration of symptoms is an important measure in determining whether to perform serology testing alongside PCR testing at time of first presentation

Line 278: What is is the clinical purpose of doing routine follow up serology testing in a PCR confirmed SARS-CoV-2 patient.

Line 317. Don’t agree with this statement. As mentioned above there have been many different reasons for doing antibody testing. Without incorporating perspectives of what is happening on the ground as things change fast, it is inappropriate to link with unreferenced and likely now updated “clinical guidance”

Line 327: data of PCR test is not a good proxy for date of symptom onset. PCR testing in many studies is reported to be done between 5 and 15 days post onset of symptoms when patients present to healthcare. Accept this may be different in the US

Reviewer #3: 1) Real-world utilization of SARS-CoV-2 serological testing in RNA positive patients across the United States

A very frustrating, slightly annoying but not unexpected finding that there is no coherent data collection between providers. That said the major quantitative finding of the study appears to be of the 930,669 individuals with positive RNA for SARS-CoV-2 identified, 15% had serotesting <14 days from the RNA positive test. Positivity at 14 days can be as high as 95% for hospitalized patients corresponding to large viral loads and arguably slow immune response.

The study was surprisingly predominantly female despite the prevalence and increased risk in males over 55. Why in such a large cohort was this bias observed?

A solid set of data analysis but I was left with a real feeling of ‘so what?’. The study was not wrong, appears competently performed but lacked truly useful data because this is not collected by the partners. No inter-platform analysis, no identification of the antibody test performed, no analysis of false negatives from the PCR test (80%), no analysis of false positive from PCR test (4%), no analysis of estimate of recording errors (1 per 1000 entries).

6. PLOS authors have the option to publish the peer review history of their article (what does this mean?). If published, this will include your full peer review and any attached files.

Reviewer #1: No

Reviewer #2: No

Reviewer #3: No

---

## [Author Response · Author response to Decision Letter 0]

19 Jan 2022

Dear Dr. Liu:

We thank you and the reviewers for your thoughtful comments on our manuscript. We appreciate the opportunity to respond and believe the revisions have improved the manuscript. Below, please find a summary of reviewer comments (bold font), followed by our responses (regular font). We look forward to hearing from you.

Regards, 

Carla Rodriguez-Watson, PhD, MPH

Director of Research

Reagan-Udall Foundation for the FDA

Review Comments to the Author

Reviewer #1: 

The authors conducted a review of electronic health records and/or claims data across six major health systems in the United States of America to assess the real-world utilisation of serological testing for SARS-CoV-2 in patients who had tested positive for SARS-CoV-2 via RNA previously. They identified 930,669 individuals and noted that approximately 15% of them underwent serological testing within 14-days of positive diagnosis via RNA.

Unfortunately the study was limited by its retrospective nature and its reliance on EHR and claims data and as such a fair amount of data is missing including race/ethnicity data, molecular and RNA test type and symptomatology. However, as the intention of this manuscript is to characterise the utilisation of serological testing for SARS-CoV-2 in a real world setting, this missing data does not compromise the integrity of the study but in fact highlights shortcomings in the health system, in terms of critical information capturing such as demographic and clinical data.

Furthermore, the manuscript also gives insights into health practices during the pandemic including non-adherence to clinical guidelines related to SARS-CoV-2 testing and the use of serological assay. This information is important as it can inform future practice and highlight the need to follow evidence based guidance during the pandemic. Notably the data is heavily skewed towards Providers B and F (> 66% of the cohort) but as the data is presented individually this is clear. The manuscript may benefit from the inclusion of a "total" column in all tables although this may be difficult to express the data due to the large amount of missing information.

Thanks for your positive feedback. As you mentioned, due to high volume of missing data in many variables adding a separate column in each table may not be feasible and informative. However, we added a column in Table 1 to show the total amount of participants in each variable. Please see pages 10-15.

Reviewer #2: 

This is an impressive dataset collected from multiple systems addressing principles of inter-operability, population coverage and linkage of laboratory results with detailed clinical metadata to address policy or public health needs. The main concern is whether the objectives set have made sufficient progress to justify publication at this point. It also would benefit from more detailed explanation of the research questions behind the objectives set (presumably by the FDA) to support presentation of the progress made in the form of a research publication, rather than alternatively as a published interim project progress report. 

Thanks for the comment. We modify the introduction to explain in more detail why we conducted this study (what are the gaps), and how it may benefit the audience of Plos One (Policy makers, clinicians etc.) Please see pages 4-6.

It also seems aligned to the requirements of the US approach to FDA diagnostics approvals, implementation and guidance. This makes it a bit hard to clearly see the specific research questions as opposed to providing data to support decision making within the US diagnostics regulatory framework. The comments below expand on these concerns which are raised from perspective of a non-US infectious diseases clinician involved in SARS-CoV-2 testing including serological tests. Consequently some concerns may be easily addressed or reflect misunderstanding due to lack of familiarity with the US setting and how this work fits in.

Thank you for your acknowledgement of the breath and complexity of the data collected and the challenge presented. We agree that the information presented in this manuscript could have been better tied to the objectives of the overall project. We have added clarity to the manuscript that this descriptive analysis to describe the use of serological tests in the U.S: who is using them, when they are being used, etc., and the availability of certain data elements (and the lack thereof), including race and manufacturer test name, is an essential first step for a future evaluation of the real-world performance of serology tests. In particular, we found that for a large number of test results, the manufacturer test name (e.g., Abbott Architect IgG) was not available. If the test name cannot be linked to the result, then the accuracy of the test cannot be determined. Standard coding (e.g., LOINC) do not capture test name. Furthermore, studies of real-world performance are needed because currently available serology tests were issued under emergency use authorization, which does not require the same evidentiary standard as normal FDA clearance for diagnostic tests. Valid serology tests are important because they provide important information about immune response in the context of evolving variants of SARS-CoV-2. 

We edited the manuscript to address your comments. Please see our responses below. 

Abstract - conclusion: how exactly does this methodological approach actually address the question of determining performance data or linking with manufacturer and clinical data. It does not seem appropriate to say the need for more efficient testing strategies without presenting the actual strategies and rationale for testing in these different settings at that time. 

We have clarified the text, particularly in the following places: lines 55-60; 68-81; lines 112-115; lines 421-426.

We also have removed the comment regarding testing strategies.

Given the dynamic nature of this disease there is potential for findings of a study like this to not be relevant by the time a follow up study is done or recommendations are made for the future. For example, in the study period serology testing was sometimes performed to help with diagnosis of what was a new condition. Antibodies appear around day 10 and so without vaccination or potential for prior infection, it was not inappropriate to perform serology to help with diagnosis. Duration of symptoms was an important clinical measure not collected in this study.

Agree that COVID, as well as the tests being put out on the market, is a dynamic situation. However, the gaps we identify in data availability have been persistent and will continue to persist, thereby limiting the effectiveness of the EUA mechanism. The EUA mechanism was essential to get much needed tests into the public. Real-world data (or observational data from healthcare interactions) offers an unprecedented opportunity to efficiently assess the real-world performance of these tests only if the data are available. Learnings from this descriptive analysis, particularly the need to integrate data, may inform our ability to identify and respond to future pandemics and emergencies. See lines 55-60; 68-81; 112-115; 365-368.

In addition, patients now get serology testing on admission to determine which SARS-CoV-2 RNA-PCT positive patients are antibody negative and might require Regeneron or similar

Finally, now that vaccination is widespread serology testing is used more to assess immunity (not prevalence/incidence of disease) or as a universal hospital admission test to inform infection control decisions (ie antibodies +/- history of vaccination = protected and less likely to acquire SARS-CoV-2).

We agree with the statements. Our purpose was to describe how serology tests are being used. We found that 15% had a serology test within 15 days of the positive RNA; and that most serology tests in that timeframe occurred on the same day as the RNA. As a critical first step to understanding the performance of serology against a known positive sample, it’s important to establish when serology tests were used in relation to the molecular test. As you suggest, we incorporated how these results may be interpreted in the context of changing clinical practice: lines 334-344.

Line 69 serology is not only done to determine prevalence estimates

Agree. We clarified the language in lines 334-344.

Line 73 Please explain for those not familiar with US regulatory processes what the relevance of 510(k) is and what the link is between approvals and this collection of real world evidence

In line with your comment, we removed the reference to 510(k).

Line 81 -83. Interoperability and linkage of tests with metadata from EHR impacts on all of medicine but what are the advantages of this approach here for determining test performance and linking with clinical symptoms. Please explain the benefits compared with studies conducted at representative institutions / different clinical settings where local laboratory and clinical data can be collected alongside guidelines and rationale. Realise the national approach will get there but it is a long and complex road to get to the granularity required to meet stated objectives.

Agree. We have edited the manuscript to reflect more clearly that this analysis was not an attempt to create a national data source, but to conduct 6 analyses in parallel to describe the robustness of findings. The purpose of the manuscript is as much to discuss differences in data available and how that may/may not impact results. Please see lines 45-46; 177-180; 190-192; 372-385.

Line 93. The FDA asked for this consortium to be established with specific purpose. The consortium has a broad range of experts and links diverse providers. Line 100 sets 3 different objectives. How does the meeting of those objectives link into a research question and publication?

Thank you for the comment. We have edited the manuscript to reflect more clearly how these objectives are a necessary first step to understand real-world performance of serology. Please see lines 55-60; 68-80; lines 113-115; lines 421-426.

Line 109: How did these organisations came together and are representative of the organisations involved in serology. They presumably were not randomly selected. The authors do say heavy representation from certain states

Great question. We have clarified how these organizations were selected for study. Please see lines 118-123.

Line 126. March to September was very early on in the pandemic. First wave. Diagnostics and guidelines have improved significantly. All tested parameters are very different now, hence making eneralizability comments, even for the US, seems tenuous for today.

Thanks for the comment. We have clarified the manuscript to describe the descriptive intent of this analysis, rather than an assessment of accuracy. These results put performance into context. The learnings about the available data and data sources are instructive for current and future pandemics. We address this in lines 55-60; 68-80; 113-115; 365-369.

 From line 128: it is hard to understand the locations and factors underpinning access to diagnostic tests from outside the US system to interpret results and assess potential biases and confounders

This comment does appear to pair with line 128. 

Line 251; Duration of symptoms is an important measure in determining whether to perform serology testing alongside PCR testing at time of first presentation

We agree with you. While we did not collect narrative information, we did collect diagnoses at presentation during the initial visit.

Line 278: What is is the clinical purpose of doing routine follow up serology testing in a PCR confirmed SARS-CoV-2 patient.

We have tried to enumerate the multiple and changing reasons to conduct serology testing among known PCR confirmed patients, as well as among unknown PCR status. Please see lines 336-344.

Line 317. Don’t agree with this statement. As mentioned above there have been many different reasons for doing antibody testing. Without incorporating perspectives of what is happening on the ground as things change fast, it is inappropriate to link with unreferenced and likely now updated “clinical guidance”

In line with your comment, we have removed language suggesting “right” or “wrong” practice and underscored that the focus of this is as a description of what is happening.

Line 327: data of PCR test is not a good proxy for date of symptom onset. PCR testing in many studies is reported to be done between 5 and 15 days post onset of symptoms when patients present to healthcare. Accept this may be different in the US.

We agree and have noted as such in lines 421-423.

Reviewer #3: 

1) Real-world utilization of SARS-CoV-2 serological testing in RNA positive patients across the United States. A very frustrating, slightly annoying but not unexpected finding that there is no coherent data collection between providers. That said the major quantitative finding of the study appears to be of the 930,669 individuals with positive RNA for SARS-CoV-2 identified, 15% had serotesting <14 days from the RNA positive test. Positivity at 14 days can be as high as 95% for hospitalized patients corresponding to large viral loads and arguably slow immune response.

The study was surprisingly predominantly female despite the prevalence and increased risk in males over 55. Why in such a large cohort was this bias observed?

Thanks for your comment. We observe that about 53% of our sample were female which although significantly larger than male, is not overwhelming majority and in line with previous large studies of women traditionally seeking health more than men. Also of note, that most of the patients came from outpatient settings. 

A solid set of data analysis but I was left with a real feeling of ‘so what?’. The study was not wrong, appears competently performed but lacked truly useful data because this is not collected by the partners. No inter-platform analysis, no identification of the antibody test performed, no analysis of false negatives from the PCR test (80%), no analysis of false positive from PCR test (4%), no analysis of estimate of recording errors (1 per 1000 entries).

Thanks for your comment. We agree with you on absence of some key elements in the current paper. However, as we mention in the limitation section, this paper is part of a larger project that answer some of those questions eventually. We decide to not put all the information in one paper for sake of practicality and clarity. Please see lines 426-430.

Besides, our focus was to describing testing patterns among those who are PCR+ as a critical first step to later assess real-world positive percent agreement between serology and active infection as assessed from +molecular test. We did not include negative cases because positive molecular test was part of inclusion criterion into cohort. Please see lines 55-60; 68-81; lines 113-115; lines 421-426.

---

## [Decision Letter · Decision Letter 1]

27 Apr 2022

PONE-D-21-30196R1Real-world utilization of SARS-CoV-2 serological testing in RNA positive patients across the United StatesPLOS ONE

Dear Dr. Rodriguez-Watson,

Thank you for submitting your manuscript to PLOS ONE. After careful consideration, we feel that it has merit but does not fully meet PLOS ONE’s publication criteria as it currently stands. Therefore, we invite you to submit a revised version of the manuscript that addresses the points raised during the review process.

We look forward to receiving your revised manuscript.

Kind regards,

Jue Liu, Ph.D.

Academic Editor

PLOS ONE

Journal Requirements:

Reviewers' comments:

Reviewer's Responses to Questions

**Comments to the Author**

1. If the authors have adequately addressed your comments raised in a previous round of review and you feel that this manuscript is now acceptable for publication, you may indicate that here to bypass the “Comments to the Author” section, enter your conflict of interest statement in the “Confidential to Editor” section, and submit your "Accept" recommendation.

Reviewer #1: (No Response)

Reviewer #2: All comments have been addressed

2. Is the manuscript technically sound, and do the data support the conclusions?

Reviewer #1: Yes

Reviewer #2: Yes

3. Has the statistical analysis been performed appropriately and rigorously? 

Reviewer #1: Yes

Reviewer #2: I Don't Know

4. Have the authors made all data underlying the findings in their manuscript fully available?

Reviewer #1: Yes

Reviewer #2: Yes

5. Is the manuscript presented in an intelligible fashion and written in standard English?

Reviewer #1: Yes

Reviewer #2: Yes

6. Review Comments to the Author

Reviewer #1: Thank you for the changes made following review. The manuscript remains relevant in that it gives insights into health practices and the use of serological testing during the early days of the pandemic. Despite rapid changes in the pandemic since the date of initial submission this information may still inform future practice with COVID and other viral infections to come.

Two minor recommendations:

Tables pages 10 - 15: Thank you for adding the total column to the table. However, the column shows a total for both "Yes" and "No" for serological testing status which adds little information. This reviewer feels it will be better to show a Total column for "Yes" and a total column for "No" in order to compare the broad outcomes of serological testing in each group.

Table, page 23: "Any Chargemaster or Medical Claim"

Please clarify in the footnotes what "Chargemaster" is. As a non-US reader/reviewer I am not sure of the relevance of this.

Thank you for resubmitting your work.

Reviewer #2: (No Response)

7. PLOS authors have the option to publish the peer review history of their article (what does this mean?). If published, this will include your full peer review and any attached files.

Reviewer #1: No

Reviewer #2: **Yes: **Jonathan Edgeworth

---

## [Author Response · Author response to Decision Letter 1]

23 Sep 2022

Reviewer Comments

Reviewer 1: 

1. Tables pages 10 - 15: Thank you for adding the total column to the table. However, the column shows a total for both "Yes" and "No" for serological testing status which adds little information. This reviewer feels it will be better to show a Total column for "Yes" and a total column for "No" in order to compare the broad outcomes of serological testing in each group.

Thank you for your comment. The comment was unclear to us. If the request is to combine all the Yes’s across all the datasets (i.e. 2191+14059+2170+2808+2137+12441), as well as No’s, that would not be appropriate because the data are not homogenous and derived from different data source types (e.g., EHRs, claims). Given the unique features of each data source (e.g., variable data availability, completeness, and characterization), this study consisted of parallel analyses (i.e., implementation of a common protocol and analysis plan in separate unique datasets) with none of the analyses representing a combined dataset.

2. Table, page 23: "Any Chargemaster or Medical Claim". Please clarify in the footnotes what "Chargemaster" is. As a non-US reader/reviewer, I am not sure of the relevance of this. 

Thank you for your comment. In the datasets, any chargemaster or medical claim refers to a comprehensive list of a hospital’s billable products, procedures, and services. We have added this description to the legend of Table 3. Please see lines 291-293. 

Reviewer 2: 

1. Has the statistical analysis been performed appropriately and rigorously?

 Thank you for the question. 

Yes, we conducted the statistical analysis appropriately and rigorously. Our objectives for this study were to 1) understand the current state of data interoperability across instrument, laboratory, and clinical data; 2) describe serological testing by demographic, environmental characteristics (e.g., geographic location), baseline clinical presentation, key comorbidities (e.g., diabetes and cardiovascular disease), and bacterial/viral co-infections (e.g., influenza), and 3) assess the timing of serology testing relative to molecular testing date by the characteristics listed above. Given these objectives, we performed a number of statistical analyses. Descriptive analyses were performed on the covariates of interest separately by each contributing data partner in accordance with a common analytic plan. Additionally, we calculated the median and interquartile range (IQR) for the number of days between RNA and the first test. Separately, we included all serology and RNA tests after the index date to describe the median and IQR for the number of molecular and serological tests conducted after the index date. A complete description of the statistical analysis that was performed can be found in lines 178-190.

---

## [Decision Letter · Decision Letter 2]

23 Jan 2023

Real-world utilization of SARS-CoV-2 serological testing in RNA positive patients across the United States

PONE-D-21-30196R2

Dear Dr. Rodriguez-Watson,

We’re pleased to inform you that your manuscript has been judged scientifically suitable for publication and will be formally accepted for publication once it meets all outstanding technical requirements.

Kind regards,

AbdulAzeez Adeyemi Anjorin, Ph.D.

Academic Editor

PLOS ONE

Additional Editor Comments (optional):

Reviewers' comments:

Reviewer's Responses to Questions

**Comments to the Author**

1. If the authors have adequately addressed your comments raised in a previous round of review and you feel that this manuscript is now acceptable for publication, you may indicate that here to bypass the “Comments to the Author” section, enter your conflict of interest statement in the “Confidential to Editor” section, and submit your "Accept" recommendation.

Reviewer #1: All comments have been addressed

2. Is the manuscript technically sound, and do the data support the conclusions?

Reviewer #1: Yes

3. Has the statistical analysis been performed appropriately and rigorously? 

Reviewer #1: Yes

4. Have the authors made all data underlying the findings in their manuscript fully available?

Reviewer #1: Yes

5. Is the manuscript presented in an intelligible fashion and written in standard English?

Reviewer #1: Yes

6. Review Comments to the Author

Reviewer #1: Thank you for the clarification and the corrections. I believe it is now acceptable for publication.

7. PLOS authors have the option to publish the peer review history of their article (what does this mean?). If published, this will include your full peer review and any attached files.

Reviewer #1: No

---

## [Editor Report · Acceptance letter]

30 Jan 2023

PONE-D-21-30196R2 

Real-world utilization of SARS-CoV-2 serological testing in RNA positive patients across the United States 

Dear Dr. Rodriguez-Watson:

I'm pleased to inform you that your manuscript has been deemed suitable for publication in PLOS ONE. Congratulations! Your manuscript is now with our production department. 

Kind regards, 

on behalf of

Dr. AbdulAzeez Adeyemi Anjorin 

Academic Editor

PLOS ONE